# HYBRID NEURAL-MPM FOR INTERACTIVE FLUID SIMULATIONS IN REAL-TIME

## ABSTRACT

We propose a neural physics system for real-time, interactive fluid simulations. Traditional physics-based methods, while accurate, are computationally intensive and suffer from latency issues. Recent machine-learning methods reduce computational costs while preserving fidelity; yet most still fail to satisfy the latency constraints for real-time use and lack support for interactive applications. To bridge this gap, we introduce a novel hybrid method that integrates numerical simulation, neural physics, and generative control. Our neural physics jointly pursues low-latency simulation and high physical fidelity by employing a fallback safeguard to classical numerical solvers. Furthermore, we develop a diffusion-based controller that is trained using a revserve modeling strategy to generate external dynamic force fields for fluid manipulation. Our system demonstrates robust performance across diverse 2D/3D scenarios, material types, and obstacle interactions, achieving real-time simulations at high frame rates ($11 \sim 29\%$ latency reduced) while enabling fluid control guided by user-friendly freehand sketches. We present a significant step towards practical, controllable, and physically plausible fluid simulations for real-time interactive applications. We promise to release both models and data upon acceptance.

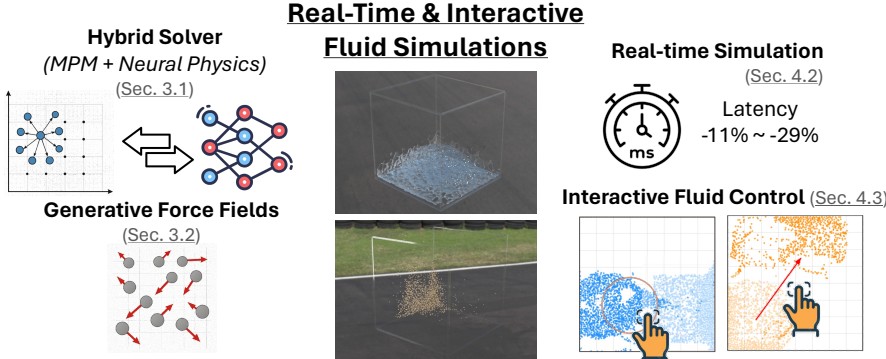

Figure 1: We target real-time, interactive fluid simulations. Our hybrid solver integrates a numerical simulator and neural physics (Section 3.1), enabling real-time simulation (Section 4.2). In addition, we generate external force fields (Section 3.2) to support users to control fluids interactively via freehand sketches (Section 4.3).

## 1 INTRODUCTION

Modeling fluid behavior is essential for advancing diverse engineering fields, including entertainment (Stam, 2023), urban planning (Blocken & Stathopoulos, 2013), fashion design (Volino et al., 2005), and virtual reality (VR) (Solmaz & Van Gerven, 2022). Moreover, controllability, aiming to instruct movements and shapes of fluids, is also a very important attribute for volumetric effects, character animations, and fluid-solid coupling (Raveendran et al., 2012). Realizing compelling and interactive physics simulations in real-time has been the long-standing objective for years in order to deliver transformative user experiences.

Traditional simulation methods, though powerful, often demand significant implementation efforts and computational costs (Bridson, 2015). Recent neural physics and machine learning approaches

present a promising path forward by learning from data, delivering transformative changes for use cases such as fluid interactions and animations (Sanchez-Gonzalez et al., 2020). However, fidelity and latency in these neural-based methods are not well-balanced. Moreover, most methods only focus on the accuracy of non-interactive applications, and their computational complexity still remains generally high for real-time scenarios (Brandstetter et al., 2022).

Motivated by the above challenges, we ask two scientific questions:

> **Q1**: *Can neural physics accelerate real-time fluid simulations and interactions?*
> **Q2**: *Can neural physics and generative methods be optimized for interactive fluid control?*

We aim to explore a novel paradigm: neural physics for interactive simulations in real-time (Figure 1). We provide affirmative answers. The core idea is to **proactively marry the strengths of numerical simulation (high fidelity), neural physics (low latency), and generative control (interactivity)** to deliver authentic and diverse fluid simulations. Specifically, neural physics is responsible for significantly low-latency fluid simulation with tolerant errors, and numerical simulation will serve as a fallback solution when fluid dynamics is increasingly complex. Furthermore, to make fluid animation compatible with user-friendly control, we introduce another diffusion-based controller to generate external force fields to assist manipulations. We summarize our contributions below:

1. We improve the **error-latency trade-off** of fluid simulation. First, to accelerate neural physics, we seek to build our graph neural network at low spatiotemporal resolution without substantial degradation in simulation accuracy (Section 3.1.1). Second, to preserve simulation fidelity and avoid error accumulation during unrolling, we make our neural physics hybrid with a safeguard condition and fallback mechanism to the classic MPM (Material Point Method) algorithm (Section 3.1.2).

2. We further aim to support **users' flexible freehand sketches** that specify desired trajectories or shapes of fluid particles to be controlled. To this end, our novel reverse simulation strategy enables the automated generation of realistic fluid control data (Section 3.2.2), which is used to train our diffusion-based generative controller (Section 3.2.3).

3. Across **diverse scenarios** (2D/3D, particle materials, presence of rigid obstacles, see Table 2), our hybrid simulator can **significantly accelerate simulations** ($11 \sim 29\%$ latency reduced) while maintaining low errors (Section 4.2), and can **control fluid particles to align with user sketches** (Section 4.3), paving the way for promising advances towards engaging interactive simulations in real-time.

## 2 BACKGROUND

We first introduce the necessary components on which our method is built, and how they can be made real-time and controllable in Section 3.

### 2.1 FLUID SIMULATIONS WITH MATERIAL POINT METHOD (MPM)

The Material Point Method (MPM) (Jiang et al., 2015; Hu et al., 2019; 2020; 2021) is a hybrid Eulerian-Lagrangian numerical technique for simulating complex interactions between solid and fluid materials, especially under large deformations and topological changes (snow, landslides, cloth, etc.). It extends the FLuids-Implicit-Particle (FLIP) (Brackbill & Ruppel, 1986) from Computational Fluid Dynamics (CFD) to solid mechanics by representing materials as a set of Lagrangian particles that carry mass, velocity ($\dot{\boldsymbol{p}}_{i,t}$), position ($\boldsymbol{p}_{i,t}$), and possible internal states. These particle quantities are first transferred to a background Eulerian grid using a particle-to-grid mapping (p2g). The equations of motion are then solved on this grid, after which updated values are mapped back to particles through grid-to-particle transfer (g2p). The particle positions ($\boldsymbol{p}$) are then advanced using the updated velocities ($\dot{\boldsymbol{p}}$), e.g., $\boldsymbol{p}_{i,t+1} = \boldsymbol{p}_{i,t} + \Delta t \cdot \dot{\boldsymbol{p}}_{i,t+1}$.

### 2.2 GNN-BASED NEURAL PHYSICS FOR PARTICLE SIMULATIONS

We denote the state of particle $i$ at time step $t$ as $\boldsymbol{x}_{i,t}$ (position $\boldsymbol{p}$, velocity $\dot{\boldsymbol{p}}$, acceleration $\ddot{\boldsymbol{p}}$, etc.), and the state of $N$ particles as $X_t = [\boldsymbol{x}_{1,t}, \ldots, \boldsymbol{x}_{N,t}]$. A *simulator* $s$ maps $T_{\text{in}}$ input states to causally

consequent future states, and can iteratively compute $X_{t_{T_{in}+1}} = s(X_{t_1}, X_{t_2}, \cdots, X_{t_{T_{in}}})$ to simulate a rollout trajectory. Following (Sanchez-Gonzalez et al., 2020), our learnable simulator $s_\theta$ adopts a particle-based representation of the physical system, which can be viewed as message-passing via a graph neural network (GNN).

**Input.** Our neural physics simulator $s_\theta$ takes the input of particle $i$ as: a sequence of 5 previous velocities (via finite differences from $T_{in} = 6$ previous locations), and features for materials (e.g., water, sand, rigid, boundary), i.e., $\boldsymbol{x}_{i,t_{k-T_{in}}:t_k} = [\dot{\boldsymbol{p}}_{i,t_{k-T_{in}+2}}, \ldots, \dot{\boldsymbol{p}}_{i,t_k}, \boldsymbol{f}_i]$ at time step $t_k$ (Figure 2).

**GNN Design.** We first build the initial graph $G^{(0)}$ by assigning a node to each particle and connecting particles as edges within a fixed "connectivity radius" $R$. The edge embeddings are learned from relative positional displacement and the magnitude $\boldsymbol{r}_{i,j} = [(\boldsymbol{p}_i - \boldsymbol{p}_j), \|\boldsymbol{p}_i - \boldsymbol{p}_j\|]$. Our neural physics consists of a stack of $L = 10$ GNN layers. The decoder predicts the per-particle acceleration, $\ddot{\boldsymbol{p}}_i$. The training loss is the particle-level $\text{RMSE}_{\ddot{\boldsymbol{p}}} \equiv \frac{1}{N} \sum_{i=1}^{N} \frac{\|\hat{\ddot{\boldsymbol{p}}}_i - \ddot{\boldsymbol{p}}_i\|_2}{\|\ddot{\boldsymbol{p}}_i\|_2}$, where $\hat{\ddot{\boldsymbol{p}}}_i$

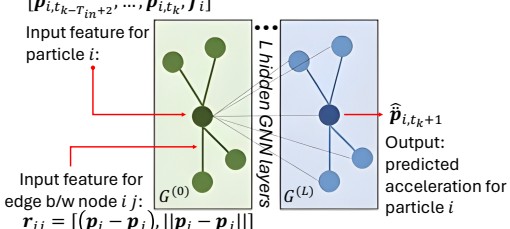

Figure 2: GNN as our neural physics simulator.

is the predicted acceleration from $s_\theta$. The future position and velocity are updated using an Euler integrator. See Appendix B for further details.

# 3 METHODS

We aim at real-time fluid simulations (Section 3.1) with interactive control (Section 3.2). Our method is overviewed in Figure 3.

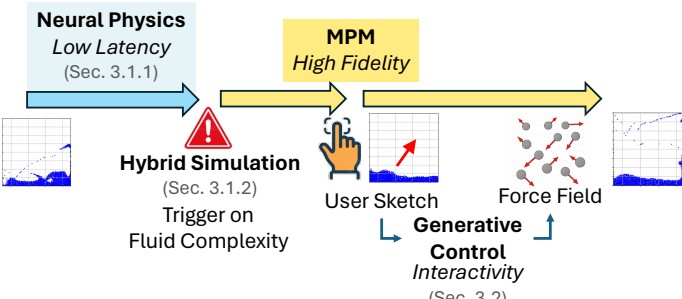

Figure 3: Method Overview. To achieve real-time simulations, we cut latency by learning neural physics at a coarse spatiotemporal resolution, while safeguarding fidelity by automatically falling back to an MPM solver when complex fluid phenomena arise (Section 3.1). For interactive control, we train a diffusion-based generative model that infers external force fields directly from user sketches (Section 3.2).

## 3.1 HYBRID REAL-TIME FLUID SIMULATION

Traditional numerical methods such as MPM provide high-fidelity simulations, but their computational cost is prohibitively high. In contrast, neural physics models can achieve significantly faster simulations by operating at low spatiotemporal resolution; however, this efficiency often comes at the cost of increased simulation errors.

**This trade-off highlights our central motivation for building hybrid simulations.** By primarily leveraging neural physics for efficient updates (Section 3.1.1) while incorporating a mechanism that falls back to MPM in challenging scenarios for simulation fidelity (Section 3.1.2), we can empirically ensure both efficiency and simulation quality. In this way, our simulator is designed as a hybrid system that effectively fuses the strengths of both approaches:

$$X_{t+1} = \begin{cases} \text{Neural Physics Update} & \text{if update is "good"} \\ \text{Fallback to MPM Update} & \text{otherwise.} \end{cases} \tag{1}$$

### 3.1.1 Learning Real-Time Neural Physics at Low Spatiotemporal Resolution

We first aim to accelerate efficient updates by neural physics, and we choose to train our model at low spatiotemporal resolution. As shown in Figure 4, we consider learning the neural physics on simulations with both a downsampled number of particles (ratio $r_p \in (0, 1)$) and also with a larger time step (i.e. coarser temporal discretization rate $r_t \in \mathbb{N}, r_t > 1$).

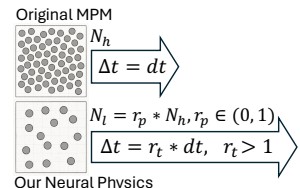

Original MPM

Our Neural Physics

Figure 4: Our neural physics accelerates simulations by learning and inferring at low spatial ($N_l$ num. particles) and temporal ($\Delta t$ time steps) resolutions, with downsampling ratios as $r_p, r_t$.

However, a key pitfall is that once the number of particles is downsampled ($N_h$ particles are merged via clustering into $N_l$, see Appendix C), we will lose the particle-wise correspondence, i.e., $\hat{\ddot{p}}_i$ ($i \in [1, N_l]$) and $\ddot{p}_j$ ($j \in [1, N_h]$) cannot align in the particle-level $\mathrm{RMSE}_{\ddot{p}}$ (Section 2.2). As a result, $\mathrm{RMSE}_{\ddot{p}}$ can no longer quantify the simulation's fidelity to the ground truth of the original spatial resolution. Inspired by Huang et al. (2021)[1], to mitigate this issue, we use a normalized grid-level $\mathrm{RMSE}_{\tilde{m}} \equiv \frac{1}{N} \sum_{i=1}^{N} \frac{\|\hat{\tilde{m}}_i - \tilde{m}_i\|_2}{\|\tilde{m}_i\|_2}$ as the evaluation metric, which essentially quantifies the mass distribution. $\tilde{m}$ is the normalized grid mass ($\tilde{m}_i = \frac{m_i}{\sum_{i=1}^{N} m_i}$) converted from particles to the grid via p2g, and $\hat{\tilde{m}}$ is the prediction by $s_\theta$. $\tilde{m}_i$ and $\hat{\tilde{m}}$ share the same grid size but can represent mass distributions from different resolutions (number of particles). A more detailed discussion regarding $\mathrm{RMSE}_{\tilde{m}}$ can be found in Appendix D. During training, we continue to optimize the surrogate loss $\mathrm{RMSE}_{\ddot{p}}$ at the low spatial resolution, thereby avoiding additional p2g operations.

In Figure 6 (a-c), we can see that by tuning spatiotemporal downsampling ratios $r_p, r_t$, we can improve the trade-off between simulation errors and latency. Based on this ablation study, we will choose $r_p = 1/1.75$ and $r_t = 2$. With this configuration, on Water 2D, we can reduce the latency of the original neural physics ($r_p = r_t = 1$) by over 78.8% (from 1.954ms to 0.4048ms).

### 3.1.2 Hybrid Simulator with Safeguard

We now discuss how to actively interact our neural physics with MPM and fuse the strengths of both approaches.

**Fluid Complexity Measures.** Intuitively, when the particle system evolves smoothly, the neural model generalizes well. In contrast, highly chaotic or abrupt behaviors, such as splashes, collisions, or multiphase interactions, often correspond to out-of-distribution (OOD) regimes, where learned models are more prone to error, suggesting that we should fall back to MPM.

We thus trigger the fallback condition based on the complexity of the current fluid dynamics being simulated by neural physics. Our choice of fallback trigger is motivated by two key considerations. *First*, this trigger should faithfully indicate the **fluid complexity**. *Second*, the safeguard should be **computationally efficient**, since we need to densely monitor them during the simulation of neural physics.

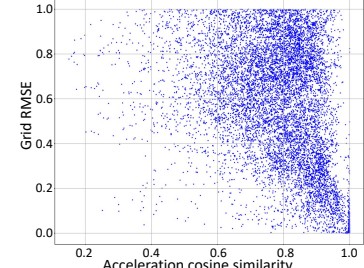

Figure 5: Negative correlation between "cosine similarity of particle accelerations over frames" vs. "simulation errors of neural physics". Scenario: Water 2D. Spearman correlation: -0.3902.

Specifically, we consider the cosine similarity of per-particle acceleration over a window of history (window size as $\delta t = 10$ steps by default): $\frac{1}{N} \sum_i^N cos(\ddot{p}_{i,t-2\delta t:t-\delta t}, \ddot{p}_{i,t-\delta t:t})$. In contrast, we also tried to monitor the divergence of particles' velocity, which is also used to quantify the quality of incompressible fluid simulations in previous works (Gao et al., 2025). However, its computation is significantly more expensive due to the use of finite difference methods, resulting in increased latency. We show the negative correlation between this cosine similarity and the neural physics simulation error in Figure 5, which indicates that whenever particles' accelerations start diverging, we should fall back to MPM.

---

[1]See Section 3.1 (page 4), paragraph "Goal and Reward" in Huang et al. (2021).

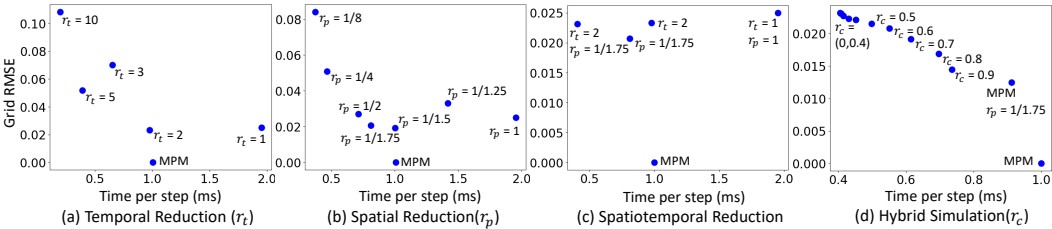

Figure 6: Ablation studies of the trade-off between grid-level $\text{RMSE}_{\tilde{m}}$ vs. simulation latency. Left to right: temporal reduction $r_t$ (train neural physics with reduced particles $N_l$), spatial reduction $r_p$ (train neural physics with larger time step $\Delta t$), spatiotemporal reduction (combine $r_t = 2$ and $r_p = 1/1.75$), and hybrid with MPM (at $r_p = 1/1.75$) with different thresholds $r_c$. Scenario: Water 2D.

**Triggering MPM by Fluid Complexity.** With our fluid complexity metric, we choose to trigger the MPM fallback mechanism by a threshold $r_c$:

$$X_{t+1} = \begin{cases} \text{Neural Physics Update} & \text{if } \frac{1}{N}\sum_i^N \cos\left(\ddot{\boldsymbol{p}}_{i,t-2\delta t:t-\delta t}, \ddot{\boldsymbol{p}}_{i,t-\delta t:t}\right) > r_c \\ \text{Fallback to MPM Update} & \text{otherwise.} \end{cases} \quad (2)$$

In Table 1, we see that when increasing our threshold $r_c$ (i.e. MPM will be more frequently triggered), the simulation fidelity will be corrected by MPM ($\text{RMSE}_{\tilde{m}}$ is improved), and the latency will increase due to heavy computations of MPM. Thus, we need to choose a threshold $r_c$ such that we can improve our trade-off between $\text{RMSE}_{\tilde{m}}$ and latency. In Figure 6 (d), we tune this threshold, and choose $r_c = 0.8$ to balance the improvements over $\text{RMSE}_{\tilde{m}}$ and latency.

**Table 1:** Grid $\text{RMSE}_{\tilde{m}}$ vs. time per step with hybrid simulations triggered by different thresholds (Water 2D).

| Threshold $r_c$ | 0.0 | 0.1 | 0.2 | 0.3 | 0.4 | 0.5 | 0.6 | 0.7 | 0.8 | 0.9 |
|---|---|---|---|---|---|---|---|---|---|---|
| Grid $\text{RMSE}_{\tilde{m}}$ | 0.0232 | 0.0230 | 0.0227 | 0.0223 | 0.0221 | 0.0215 | 0.0208 | 0.0192 | 0.0169 | 0.0144 |
| Time per step (ms) | 0.4048 | 0.4081 | 0.4147 | 0.4301 | 0.4516 | 0.4977 | 0.5509 | 0.6137 | 0.6966 | 0.7356 |

We finalize our hybrid solver using this threshold. In Figure 7, we demonstrate trajectories simulated by the original neural physics and our hybrid solver (from the same initial condition, of the same number of steps $T$). Although the original neural physics ($r_p = r_t = 1$) shows lower rollout errors in the early stage (black curve, due to simulation at high resolution), it quickly accumulates long-term errors. In contrast, after triggering the fallback to MPM (yellow areas), our error is suppressed and we finish the simulation much faster. Thus, our hybrid solver improves both rollout $\text{RMSE}_{\tilde{m}}$ and latency.

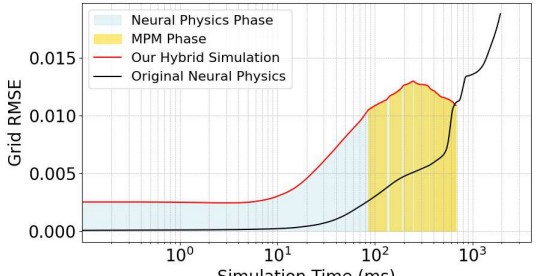

Figure 7: Error trajectories during simulation (Water 2D). Simulating the same number of steps ($T = 1000$), our hybrid solver takes significantly less time (676.4ms) than the original neural physics (1931.1ms), and the final error is also reduced (grid $\text{RMSE}_{\tilde{m}}$) (0.0109 vs. 0.0188).

### 3.2 Interactive Fluid Control by Generating Dynamic Force Fields

#### 3.2.1 Motivations

Fluid control is essential in computer graphics, where liquid animations convey expressive, story-driven scenes and key visual ideas like splash shapes or motion (Yan et al., 2020). Manual fluid control produces unnatural effects and forces artists to rely on slow, trial-and-error methods (Pan et al., 2013). This underscores the need for intuitive tools that let users shape visuals directly, without complex physics. Yet, achieving the desired appearance of fluid control remains difficult. Fluid dynamics are intrinsically chaotic and unpredictable. Setup and tuning of fluid control is tedious and repetitive. Moreover, recording real fluid motion is also expensive and hard to customize.

In our paper, we mainly consider the following use case: during a fluid simulation, a user would like to draw a simple sketch and provide it as a control signal, following which the fluid particles should move (Figure 8 bottom panel). However, how to *artistically* manipulate fluid particles to follow the user's sketch should be automatically designed by our system.

**Why Generative Fluid Control via Diffusion Models?** Diffusion-based generative models are a natural choice for fluid control because they combine strong conditional generation with temporal coherence and spatial flexibility. Recent works show that diffusion can precisely steer motions when conditioned on trajectories or region cues in video (Wang et al., 2024b; Zhang et al., 2024; Yang et al., 2024) and predict spatiotemporal physics dynamics with high fidelity (Hu et al., 2024; Zhou et al., 2024; Dong et al., 2025; Kohl et al., 2023). This suggests that **diffusion models can map sketch-like intent to physically meaningful velocity or force fields over time.** In fact, recent works have already learn to synthesize fluid velocities with deep generators (Chu et al., 2021; Yan et al., 2020). In contrast, traditional control pipelines in computer graphics often require bespoke assets or costly optimization, e.g., precomputed templates or offline heuristics (Schoentgen et al., 2020; Pan et al., 2013), which limits interactivity and generalization.

### 3.2.2 DATA GENERATION VIA REVERSED SIMULATION

The key to training a generative model for our fluid control is to automatically collect training data in principle. Specifically, we have two highly nontrivial sub-tasks: 1) Design a large number of diverse scenarios of fluid particles with artistic control effects (i.e. fluid particles move along a desired direction or fill a pre-defined shape, in an organized instead of a chaotic manner); 2) Solve a spatiotemporal external force field that will be applied to the particles, such that the artistic control effect can be fulfilled driven by the composition of gravity, particle interactions, and the proposed force field.

We address these challenges with a reverse simulation strategy. The core idea is to **solve the required force fields** that can **reverse the fluid dynamics** of artistic effects. We have the following steps:

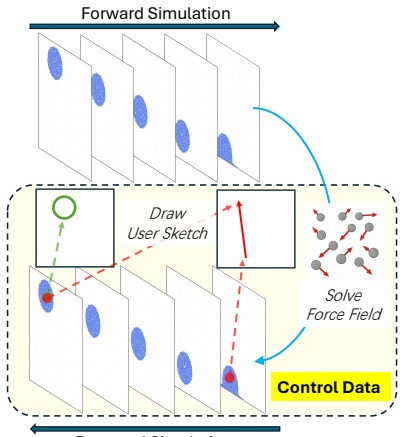

Figure 8: We prepare training data for generative control via solving external force fields that can reverse a forward simulation. We also prepare user sketches (arrow, ellipse) that depict movements or target shapes of particles (implementations in Appendix C).

**1) Forward Simulation.** We randomly simulate a trajectory of fluid dynamics $\boldsymbol{X} = (X_1, X_2, \cdots, X_{T_{ctr}})$, with different initial conditions (positions or velocities of particles).

**2) Reversed Simulation.** We iteratively solve the required acceleration[2] $\boldsymbol{a}_t$ that can restore *positions* of each fluid particle reversely, from $X_{T_{ctr}}$ to $X_1$:

$$(\boldsymbol{a}_t + \boldsymbol{g})\Delta t = \frac{\boldsymbol{p}_{t-1} - \boldsymbol{p}_t}{\Delta t} - \dot{\boldsymbol{p}}_t$$
$$\boldsymbol{a}_t = \frac{(\boldsymbol{p}_{t-1} - \boldsymbol{p}_t) - \dot{\boldsymbol{p}}_t \cdot \Delta t}{(\Delta t)^2} - \boldsymbol{g}. \tag{3}$$

Equation 3 stems from the discretized second-order difference equation of motion, and provides a physically interpretable approximation of the acceleration needed to move from $\boldsymbol{p}_t$ to $\boldsymbol{p}_{t-1}$, subtracting out the known gravitational acceleration $\boldsymbol{g}$. Our target force field can be non-linear (see Fig. 13 in our Appendix for a simulation example). Moreover, in Fig. 15, the cases where fluid shapes change during control indicate that our diffusion model is trained to predict these non-linear force fields.

**3) Generation of Control Sketches.** Finally, based on $\boldsymbol{X}$, we generate the user's sketch that depicts the general movements of particles. We support both directional arrows for movement guidance and one-stroke freehand oval shapes to indicate target regions, as shown in Figure 8. See our Appendix C for details of implementing freehand arrows and oval shapes. Note that in 3D scenarios, we use the arrow width to indicate depth (Pan et al., 2013).

For simplicity, we will by default control the fluid particles for 100 MPM steps ($T_{ctr} = 100$). That means all our control trajectories will have 100 steps. While it is possible to employ dynamic neural architectures (Yu et al., 2018) to adaptively adjust the number of MPM steps for this control based on the control complexity, we leave it as a future work.

---

[2]Equivalently, the force field if all particles have the same constant mass

### 3.2.3 DIFFUSION-BASED FLUID CONTROLNET

Conditioned on previous particle trajectories and a user's sketch, our diffusion-based Fluid ControlNet will predict a dynamic force field ($a$ in Equation 3). We control fluid particles by applying this predicted force field atop MPM, since MPM can explicitly take external forces and update the velocity advection.

Our design follows standard practice in controllable generation (Zhang et al., 2023; Wang et al., 2024b). We accept user sketches and extract features for spatially grounded conditioning, enabling free-form arrows, regions, and shapes.

We show our architecture design in Figure 9. Specifically:

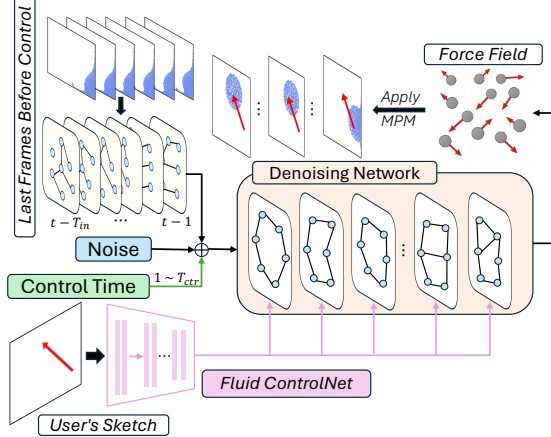

Figure 9: Architecture design of our Fluid ControlNet.

① Our diffusion-based Fluid ControlNet shares the same backbone and input particle features as our neural physics (Section 2.2).

② Parallel to the backbone, we extract the embeddings of the user's sketch input using a convolutional neural network (CNN) and concatenate them with the diffusion timestep embeddings to guide the generation process. We also embed the current control time step into a latent space and integrate it into the initial noise.

③ The training target will be the ground truth force fields we simulate in Section 3.2.2.

④ During training, noise is added to the ground-truth force (or acceleration) field, which is then used as input. The diffusion model's inputs are: (1) the previous 6 positions and velocities, (2) the noisy force field (Gaussian noise during inference), and (3) the embedding of the control timestep.

⑤ The output of our Fluid ControlNet is an external force field that will be applied to particles during MPM simulations.

⑥ Along the MPM simulation, our Fluid ControlNet will unroll the subsequent temporal force fields.

See Appendix C for details of the architecture of our Fluid ControlNet.

## 4 EXPERIMENTS

### 4.1 SETTINGS

**Physical Domains and Simulations.**  To build our hybrid simulator, we prepare our own ground truth simulations with the Taichi package (Hu et al., 2019; 2020; 2021) on GPUs, with settings closely aligned with (Sanchez-Gonzalez et al., 2020). Different simulation scenarios are summarized in Table 2. We include diverse initial conditions (position, velocity) and numbers of particles. We fix our grid size as $128 \times 128$ for 2D and $64 \times 64 \times 64$ for 3D, and use time step $dt = 2.5$ms in simulations.

Table 2: Datasets. $N_h$: Max number of particles at the original spatial resolution. $T$: total time steps. $M$: number of simulation trajectories.

| Domain | $N_h$ | $T$ | $M$ |
|---|---|---|---|
| Water (2D) | 4k | 1k | 1k |
| WaterRamps (2D) | 3.3k | 600 | 1k |
| Sand (2D) | 4k | 320 | 1k |
| SandRamps (2D) | 3.3k | 400 | 1k |
| Water (3D) | 4k | 800 | 1k |
| Sand (3D) | 4k | 350 | 1k |
| Water-Sand (2D) | 4k | 500 | 1k |

For different simulation scenarios, we train separate neural physics models and Fluid ControlNet. This design choice follows prior work (Sanchez-Gonzalez et al., 2020) where a dedicated model is also trained per scene to better capture scene-specific dynamics.

**Evaluation.**  To report quantitative results, we evaluated our models by computing **rollout** metrics on held-out test trajectories, drawn from the same distribution of initial conditions used for training. As discussed in Section 3.1.1, we use grid-level $\text{RMSE}_{\bar{m}}$ to compare predictions at lower spatial resolution with the original ground truth.

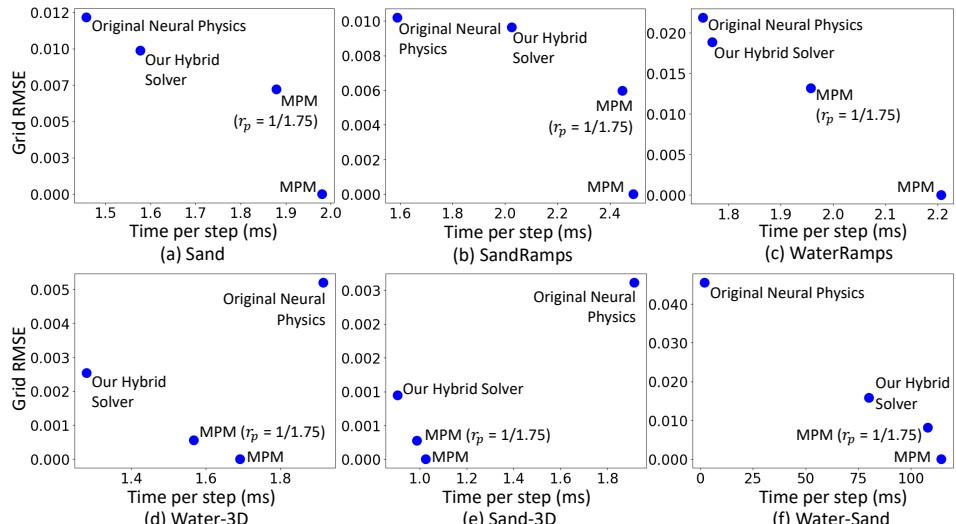

Figure 10: Trade-off between simulation error (grid $\text{RMSE}_{\tilde{m}}$) and latency, comparing different methods. (a) Sand (2D); (b) SandRamps (2D); (c) WaterRamps (2D); (d) Water (3D); (e) Sand (3D); (f) Water-Sand (2D). Overall, our hybrid solver achieves a balanced trade-off between RMSE and simulation latency, outperforming both neural physics and MPM.

### 4.2 FLUID SIMULATION ACCELERATION

Our hybrid simulator can consistently achieve real-time fluid simulations with preserved simulation fidelity across both 2D and 3D cases. We show the trade-off between simulation error and latency in Figure 10, where we compare our hybrid solver with the original neural physics ($r_p = r_t = 1$) (Sanchez-Gonzalez et al., 2020), MPM (Hu et al., 2019; 2020; 2021), and another MPM that also simulates at low spatial resolution ($r_p = 1/1.75$). On 2D scenarios, our hybrid solver consistently balances the neural physics and MPM, achieving both reduced simulation latency and preserved simulation errors. For example, on multiple materials (Water-Sand 2D), our hybrid solver can accelerate MPM from 0.114s per frame to 0.08s, with a 29.8% reduction. On 3D, the neural physics at $r_p = r_t = 1$ is extremely slow, whereas our hybrid solver improves both latency and errors. For example, on Sand 3D, we reduce the latency of MPM by 11.8%, from 1.02ms to 0.90ms. Additionally, we compare with other previous methods in Appendix E.

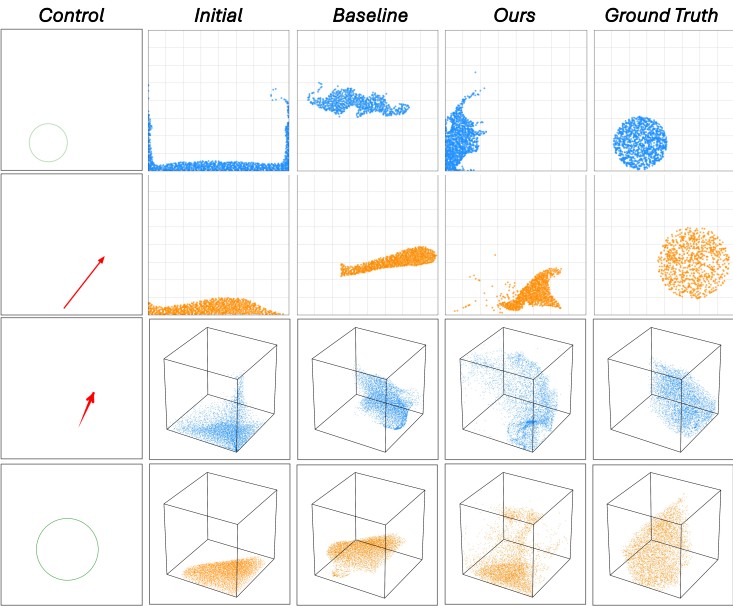

Figure 11: Visualization of generative fluid control. Rows from top to bottom: Water (2D), Sand (2D), Water (3D), Sand (3D).

### 4.3 GENERATIVE FLUID CONTROL

We show visualizations of our generative fluid control in Figure 11. We compare with a baseline, where particles are controlled with a spatiotemporal constant force field, with the force magnitude and orientation solved by moving particles from $X_{T_{\text{ctr}}}$ to $X_1$.

Table 3: Grid $\text{RMSE}_{\tilde{m}}$ between ground truth and predictions at the last time during fluid control.

| Method | Water (2D) | Sand (2D) | Water (3D) | Sand (3D) |
|---|---|---|---|---|
| Baseline | 0.0908 | 0.1151 | 0.0019 | 0.0022 |
| Ours | 0.0802 | 0.0924 | 0.0013 | 0.0019 |

We also quantitatively evaluate the control in Table 3, where we calculate the grid-level $\text{RMSE}_{\tilde{m}}$ between the ground truth and the prediction at the last time step, since our main concern is the recovery of the shape of the ground truth at the end of the simulation. In sum, we can see that our diffusion-based Fluid ControlNet can move particles to better align with the user sketches.

### 4.4 COMPLETE RESULTS: HYBRID SIMULATION + FLUID CONTROL

Finally, we present the result from our complete pipeline in Figure 12. Particles are first simulated by our hybrid solver, where we start with the neural physics (at low spatiotemporal resolution) and is triggered to MPM once the fluid complex is high. Then, a user draws a sketch to control, and our diffusion-based Fluid ControlNet takes both this sketch and recent particle states as inputs, and generates external force fields to control particles.

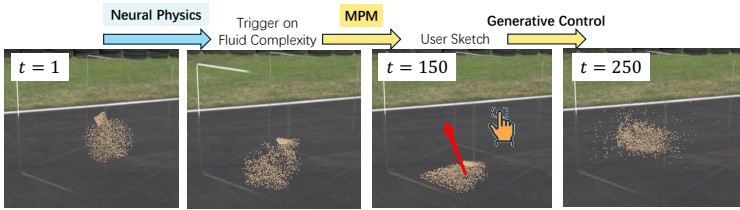

Figure 12: Complete results: hybrid simulation + fluid control. We start the simulation with our neural physics, which is then triggered to MPM. At $t = 150$, a user presents the control sketch.

## 5 RELATED WORKS

Our work builds upon recent advances in three key areas: **fluid modeling and animation**, **fluid control**, and **controllable video generation**. Research in fluid simulation has progressed from graph-based models like GNS to scalable, physics-informed hybrid solvers such as Neural SPH and MPMNet, which balance accuracy and performance (Sanchez-Gonzalez et al., 2020; Li et al., 2023; Toshev et al., 2024). Concurrently, fluid control has shifted from costly optimization towards artist-friendly, generative methods that use sketches or templates to direct fluid behavior (Yan et al., 2020; Chu et al., 2021; Schoentgen et al., 2020). We are also inspired by recent progress in controllable video generation, where diffusion models now allow for fine-grained, disentangled control over object and camera motion (Yin et al., 2023; Wang et al., 2024b; Zhang et al., 2024). A detailed discussion of these related works is provided in Appendix A. In our work, we introduce a hybrid neural-numerical framework that generates accelerated, high-fidelity, controllable fluid simulations from freehand sketches by leveraging the neural-numerical simulator and diffusion-based generative controller.

## 6 CONCLUSION

In this work, we introduced a novel hybrid neural physics framework that effectively bridges the gap between high-fidelity physical simulation and real-time interactive control. By combining learned graph-based neural simulators with a fallback to classical MPM solvers, we achieved robust, low-latency fluid dynamics capable of handling complex scenarios without sacrificing accuracy. Additionally, we developed a diffusion-based generative controller trained via reverse modeling, enabling intuitive user interaction through freehand sketches for dynamic fluid control. Extensive experiments across 2D and 3D domains demonstrate that our approach not only accelerates fluid simulations but also provides controllable and physically plausible outcomes. This hybrid paradigm represents a step forward in making real-time, artist-friendly fluid simulation practical for applications in graphics, design, and virtual environments.

THE USE OF LARGE LANGUAGE MODELS (LLMS)

LLMs did not play a significant role in either the research ideation or the writing of this paper. Their use was limited to correcting minor grammatical issues and typographical errors.

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

## A    DETAILS OF RELATED WORKS

**Fluid Modeling and Animation**    Learning-based fluid simulators have progressed from graph-based models to hybrid, physics-informed approaches. DPI-Net (Li et al., 2018) introduced dynamic interaction graphs with hierarchical message passing to model interactions across particles. This was unified in GNS (Sanchez-Gonzalez et al., 2020; Pfaff et al., 2020; Kumar & Vantassel, 2022; Kumar & Choi, 2023), enabling generalized simulation of fluids, solids, and deformables. Hybrid solvers like MPMNet (Li et al., 2023) and NeuralMPM (Rochman-Sharabi et al., 2024) adopt the Material Point Method for scalability. Neural SPH (Toshev et al., 2024) integrates SPH priors to stabilize rollouts, while NeuroFluid (Guan et al., 2022) combines learned dynamics and rendering from videos. These advances balance physical accuracy with real-time performance. In our work, we propose a hybrid approach that combines neural and numerical methods to enable accelerated and high-fidelity fluid simulation.

**Fluid Control**    Recent work in fluid control aims to make simulation more intuitive and accessible. Traditional methods using space-time optimization were costly and hard to tune. Yan et al.(Yan et al., 2020) addressed this with a sketch-based system using conditional GANs to generate liquid splashes. Pan et al.(Pan et al., 2013) enabled interactive control through sketching and mesh dragging. Chu et al.(Chu et al., 2021) used GANs to infer fluid motion from static fields with semantically controllable features. Schoentgen et al.(Schoentgen et al., 2020) introduced reusable templates for particle-based animations. These approaches shift toward flexible, artist-friendly tools. We tackle the case where only a freehand sketch is given, and the generative controller is tasked with producing the intended artistic fluid behavior.

**Controllable Video Generation**    Controllable video generation has advanced rapidly with diffusion models, especially in disentangling motion control. DragNUWA (Yin et al., 2023) enabled trajectory-based editing, while MotionCtrl (Wang et al., 2024b) and Direct-a-Video (Yang et al., 2024) decoupled camera and object motion. CameraCtrl (He et al., 2024) and CamCo (Xu et al., 2024) refined camera control using geometric cues. MotionDirector (Zhao et al., 2024) and Boximator (Wang et al., 2024a) allowed user-customized motion, and SparseCtrl (Guo et al., 2024) enabled sparse, entity-level conditioning. Tora (Zhang et al., 2024) unified text, image, and trajectory inputs for physics-aware generation. Inspired by these approaches, we leverage forward simulations and compute control forces via reversed simulation.

## B    DETAILS OF NEURAL PHYSICS SIMULATOR

### B.1    PARTICLE SIMULATIONS AS MESSAGE-PASSING ON A GRAPH

We denote the state of a particle $i$ at time step $t$ as $\boldsymbol{x}_{i,t} \in \mathbb{R}^D$, and the collective state of $N$ particles as $X_t = [\boldsymbol{x}_{1,t}, \ldots, \boldsymbol{x}_{N,t}] \in \mathbb{R}^{N \times D}$. Applying physical dynamics over multiple timesteps yields a trajectory of particle states, $\boldsymbol{X}_{t_1:t_{T_{\text{in}}}} = [X_{t_1}, X_{t_2}, \cdots, X_{t_{T_{\text{in}}}}] \in \mathbb{R}^{T_{\text{in}} \times N \times D}$. In essence, the simulator $s : \mathbb{R}^{T_{\text{in}} \times N \times D} \to \mathbb{R}^{N \times d}$ ($d = 2$ or 3 for 2D/3D) leverages the current $T_{\text{in}}$ particle states as input to predict their future motion, capturing the underlying dynamics using methods ranging from simple Euler integration to advanced numerical or data-driven techniques. If a simulator is learnable, it can be represented as $s_\theta$, a parameterized function approximator. The simulator then iteratively computes future states, such as $\tilde{X}_{t_{T_{\text{in}}+1}} = s(\tilde{X}_{t_1}, \tilde{X}_{t_2}, \cdots, \tilde{X}_{t_{T_{\text{in}}}})$, where each newly predicted state is appended to simulate a rollout trajectory over time.

Our learnable simulator $s_\theta$ represents the physical system as interacting particles, where dynamics emerge from exchanges of energy and momentum with neighbors. To ensure robust simulation quality, $s_\theta$ must generalize across diverse interaction patterns and physical scenarios. This particle-based approach naturally maps to message passing on a graph, with particles as nodes and pairwise interactions as edges, making graph neural networks (GNNs) a suitable modeling choice.

## B.2 Details of Graph-based Neural Physics

Following (Sanchez-Gonzalez et al., 2020), we implement our neural physics with GNN, and use standard nearest neighbor algorithms (Dong et al., 2011; Chen et al., 2009; Tang et al., 2016) to construct the graph.

**Input.** In our learnable simulator $s_\theta$, the input state vector for each particle $i$ at time step $t_k$ includes a sequence of 5 previous velocities (via finite differences from $T_{\text{in}} = 6$ previous locations), and static features representing material properties (e.g., water, sand, rigid, boundary particle). In practice, only the position vectors $\boldsymbol{p}_i$ are stored in our datasets; the velocities $\dot{\boldsymbol{p}}_i$ and accelerations $\ddot{\boldsymbol{p}}_i$ are computed on the fly using finite differences when needed. Formally, the node feature is defined as

$$\boldsymbol{x}_{i,t_{k-T_{\text{in}}}:t_k} = [\dot{\boldsymbol{p}}_{i,t_{k-T_{\text{in}}+2}}, \ldots, \dot{\boldsymbol{p}}_{i,t_k}, \boldsymbol{f}_i] \in \mathbb{R}^D,$$

where $\boldsymbol{f}_i$ denotes the concatenated material-specific features and scene boundary indicators. Specifically, the dimension of the encoded node feature vector is $D = 30$ for 2D simulations (5 2-dim velocities by finite differences, i.e., $5 \times 2 = 10$; 4 distances from the boundary; 16-dim embedding for the particle type), or $D = 37$ for 3D simulations (5 3-dim velocities by finite differences, i.e. $5 \times 3 = 15$; 6 distances from the boundary; 16-dim embedding for the particle type). See Figure 2 for an illustration. It is important to note that in our Fluid Controlnet (Section 3.2.3), the input feature dimension $D$ will increase by 16, where we embed the current control timestep into the latent space with another 2-layer MLP with SiLU activation.

To obtain more informative edge features $\boldsymbol{r}_{i,j}$, we use the relative positional displacement between a pair of adjacent particles $i$ and $j$, along with its magnitude:

$$\boldsymbol{r}_{i,j} = [(\boldsymbol{p}_i - \boldsymbol{p}_j), \|\boldsymbol{p}_i - \boldsymbol{p}_j\|].$$

Edges are added between particles that lie within a predefined *connectivity radius* $R = 0.015$, which captures local particle interactions. $R$ is kept constant for all 2D scenarios. In different 3D scenarios, a larger radius can be used to accommodate higher-resolution environments. Although $R$ is fixed in simulations, edges in the graph are still dynamically updated by comparing the current particle-wise distances to $R$. For full details of these input and target features, we refer readers to (Sanchez-Gonzalez et al., 2020).

The ENCODER $: \mathbb{R}^{N \times D} \rightarrow \mathcal{G}$ embeds particle-based states, it can be formulated as: $G^{(0)} = (\boldsymbol{V}^{(0)}, \boldsymbol{E}^{(0)}) = \text{ENCODER}(\boldsymbol{X}, \boldsymbol{r}_{i,j})$ The node embeddings $\boldsymbol{V}^{(0)} = \text{ENCODER}_V(\boldsymbol{X})$ are learned functions of the particles' states. The edge embeddings, $\boldsymbol{E}_{i,j}^{(0)} = \text{ENCODER}_E(\boldsymbol{r}_{i,j})$, are learned functions of the pairwise properties of the corresponding particles. We implement $\text{ENCODER}_V$ and $\text{ENCODER}_E$ as multilayer perceptrons (MLP), which encode node features and edge features into the latent vectors, $\boldsymbol{V}_i$ and $\boldsymbol{E}_{i,j}$, of size 128.

The PROCESSOR $: \mathcal{G} \rightarrow \mathcal{G}$ computes interactions among nodes through $L$ steps of learned message passing and outputs the final graph, $G^{(L)} = \text{PROCESSOR}(G^{(0)})$. Message passing enables information propagation among particles. Our PROCESSOR consists of a stack of $L = 10$ GNN layers, each using separate (non-shared) MLPs for updating node and edge features, along with residual connections between the input and output latent attributes of both nodes and edges. For the Fluid ControlNet setting, an additional MLP layer is used to encode the diffusion timestep and control image features; see Appendix C.2 for details.

The DECODER $: \mathcal{G} \rightarrow \mathbb{R}^{N \times d}$ extracts dynamics information (of the future state) from the nodes of the final latent graph, $\hat{X} = \text{DECODER}(\boldsymbol{V}^{(L)})$. Our DECODER is an MLP that outputs accelerations $\ddot{\boldsymbol{p}}_i$. The future position and velocity are updated using an Euler integrator.

All MLPs in PROCESSORhave two hidden layers with ReLU, followed by an output layer without activation, with a width of 128. All MLPs are followed by a LayerNorm (Ba et al., 2016).

## C  Implementations

### C.1  Latency Measurements

**Latency of Neural Physics.** We utilize the `TensorRT` library to convert the `PyTorch` model into an `ONNX` model to accelerate model inference and align it with the acceleration of MPM on

the `Taichi` kernel. However, since `TensorRT` does not support the `aggregation` operation in GNNs (i.e., aggregating information from edges to adjacent nodes), when measuring the latency, we approximate the time cost of this `aggregation` operation with a matrix multiplication between an adjacency matrix $A \in \mathbb{R}^{N \times N}$ (where $N$ denotes the number of nodes, i.e. particles), and node features ($o$), such that the aggregation becomes $A \cdot o$. All reported latency measurements are based on the median number of nodes across different scenarios in our test datasets.

**Latency of Taichi.** To enable a fair comparison under MPM simulation setting, we applied a matching latency reduction strategy to the `Taichi` implementation by skipping non-essential overhead. Specifically, we excluded the time spent on initializing the MPM state (initial positions and velocities of particles) and the cost of initializing the `Taichi` kernel at the beginning of the simulation. As a result, our comparison focuses solely on the runtime per simulation step after the `CUDA` or `Taichi` kernel has been initialized.

## C.2 Design of Fluid ControlNet

In our Fluid ControlNet, the control signal $\mathcal{C} \in \mathbb{R}^{H \times W \times 3}$ is encoded using our Fluid ControlNet. The encoded embedding is then injected into the graph-based diffusion model to guide the generation of the external field of accelerations. The Fluid ControlNet consists of 8 convolutional layers and 3 downsampling operations. It extracts multi-scale features from the control signal $\mathcal{C}$, projects each scale to a different dimensional space, and then concatenates the projected features into control embedding representation of dimension size 44. The resulting embedding is then integrated into the Processor module of the graph-based diffusion model. Notably, to better condition the diffusion process on the control signal, we draw inspiration from DiT (Peebles & Xie, 2023) and concatenate the embedding of the control signal to the diffusion time step embedding. This design choice ensures that the control condition is effectively incorporated at each diffusion step, thereby generating high-fidelity acceleration fields that can align fluid particles to the target motion or shape.

## C.3 Training

Following (Sanchez-Gonzalez et al., 2020), we normalize the input velocity to the GNNs, and apply random noises to input positions ($p_{t_1:t_{T_{in}}}$) during training. For both neural physics and Fluid ControlNet, we train with the Adam optimizer and a learning rate at $1 \times 10^{-4}$ with exponential decay. Our training batch size is 1, and we train for 2 million gradient descent steps.

**Table 4:** Training Costs (GPU hours) across different scenarios.

| GPU Hours | Water (2D) | Sand (2D) | Water (3D) | Sand (3D) |
|---|---|---|---|---|
| Neural Physics (Section 3.1) | 17.27h | 17.94h | 19.71h | 19.67h |
| Fluid ControlNet (Section 3.2) | 69.87h | 76.36h | 184.12h | 151.03h |

We include our training costs in Table 4. Neural physics requires approximately one day on a single NVIDIA 4090 GPU. For the Fluid ControlNet, training takes around three days for 2D scenarios on a single NVIDIA 4090 and six days for 3D scenarios on a single NVIDIA A40. We train both neural physics and Fluid ControlNet with the particle-level RMSE loss on predicted accelerations $\text{RMSE}_{\ddot{p}} \equiv \frac{1}{N} \sum_{i=1}^{N} \frac{\|\hat{\ddot{p}}_i - \ddot{p}_i\|_2}{\|\ddot{p}_i\|_2}$, which was defined in Section 2.2.

## C.4 Generating Users' Freehand Sketches (Arrows and Oval Shapes)

Arrows are computed by connecting the centroid ($\bar{p} = \frac{1}{N} \sum_{i=1}^{N} p_i$) of fluid particles at $t = 1$ ($\bar{p}_1$) and $t = T_{ctr}$ ($\bar{p}_{T_{ctr}}$). Based on the mean displacement vector $\Delta\bar{p} = \bar{p}_1 - \bar{p}_{T_{ctr}}$, we derive the arrow length $\|\Delta\bar{p}\|$ and orientation $\theta = \tan^{-1}(\Delta\bar{p}_y/\Delta\bar{p}_x)$. In 3D, we use the arrow width to indicate depth (Pan et al., 2013). A multi-segment arrow with varying line width is implemented as $n$-segment polyline with width modulation, where each segment's width $w_i$ ($i \in [1, n]$) is $w_i = w_{\min} + (w_{\max} - w_{\min}) \cdot \frac{\Delta\bar{p}_{z,i} - \Delta\bar{p}_{z,\min}}{\Delta\bar{p}_{z,\max} - \Delta\bar{p}_{z,\min}}$. The arrowhead adopts perspective-correct scaling.

For 2D oval sketches, shapes of particles at $t = T_{\mathrm{ctr}}$ are represented as elliptical outlines centered at $\bar{\boldsymbol{p}}_{T_{\mathrm{ctr}}}$, with radii corresponding to $\pm 2\boldsymbol{\sigma}$, where $\boldsymbol{\sigma}$ is the standard deviation of particle positions along each principal axis. This statistically-grounded ellipse captures approximately 95% particles' positions while being visually simple. Meanwhile, 2D oval-shaped control sketches can indeed be ambiguous in 3D, since it is infeasible to depict 3D volumes with a simple one-stroke 2D sketch.

### C.5 Enforcing Smoothness on Target Accelerations

We observe that ground truth accelerations solved by Equation 3 are typically complicated (see the temporal-wise cosine similarity in Figure 13), which will be challenging to learn. We thus further enforce a certain level of smoothness of the acceleration across temporal steps:

$$\ddot{\boldsymbol{a}}_{t,\mathrm{smooth}} = \ddot{\boldsymbol{a}}_t - \lambda \cdot \exp\left(-\beta \cdot \frac{\ddot{\boldsymbol{a}}_t \cdot \ddot{\boldsymbol{a}}_{t+1}}{\|\ddot{\boldsymbol{a}}_t\| \cdot \|\ddot{\boldsymbol{a}}_{t+1}\|}\right) \cdot (\ddot{\boldsymbol{a}}_t - \ddot{\boldsymbol{a}}_{t+1}) \tag{4}$$

Essentially, Equation 4 enforce decoupled smoothness over the magnitude and the orientation of accelerations over temporal steps. We choose $\lambda = 0.1$ and $\beta = 2$ in our work.

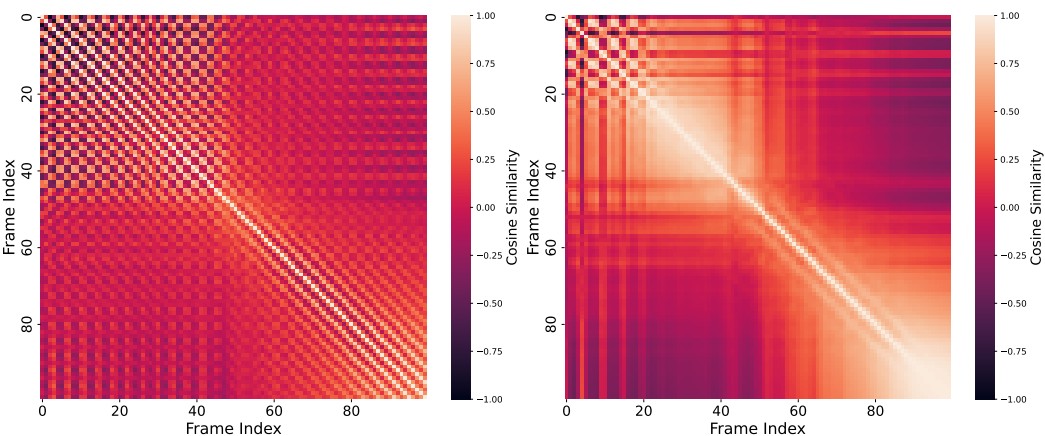

Figure 13: Step-wise correlations of ground-truth accelerations for fluid control. Left: before enforcing smoothness; Right: after enforcing smoothness.

## D More Explanation on Grid-RMSE

### D.1 Explanation of Grid-RMSE

Grid-RMSE ($\mathrm{RMSE_m}$) is a normalized error metric used to evaluate the difference between the predicted mass grid and the ground-truth mass grid in simulations. It is defined as the normalized grid-level $\mathrm{RMSE}_{\tilde{\boldsymbol{m}}} \equiv \frac{1}{N}\sum_{i=1}^{N} \frac{\|\hat{\tilde{\boldsymbol{m}}}_i - \tilde{\boldsymbol{m}}_i\|_2}{\|\tilde{\boldsymbol{m}}_i\|_2}$, which essentially quantifies the mass distribution. $\tilde{\boldsymbol{m}}$ is the normalized grid mass ($\tilde{\boldsymbol{m}}_i = \frac{\boldsymbol{m}_i}{\sum_{i=1}^{N} \boldsymbol{m}_i}$).

### D.2 The Effectiveness of Grid-RMSE

**Why we choose grid-RMSE as the metric?** The main reason we use a mass-based RMSE is that in low-resolution settings, directly computing a standard particle-level RMSE is not feasible due to significant differences in the number of particles. Moreover, performing upsampling or downsampling will break the particle alignments: the predicted and ground-truth particles will no longer be one-to-one mappings, leaving particle-RMSE infeasible.

Instead, evaluating the normalized mass distribution on the grid offers a more stable and meaningful approximation of the overall fluid shape. Essentially, it quantifies the IoU (intersection-over-union) of the predicted and target fluids.

To further validate the reliability of this metric, we have added an additional experiment based on Figure 6(d), where we compute RMSE directly at the original resolution ($r_p = 1$). As shown in

Table 5: Comparison of RMSE and mass-based RMSE ($\text{RMSE}_m$) under different $r_c$ values for the WATER2D setup ($r_t = 2, r_p = 1$).

|  | $r_c = 0$ | $r_c = 0.3$ | $r_c = 0.5$ | $r_c = 0.6$ | $r_c = 0.7$ | $r_c = 0.8$ | $r_c = 0.9$ |
|---|---|---|---|---|---|---|---|
| Particle RMSE | 0.2780 | 0.2741 | 0.2619 | 0.2528 | 0.2437 | 0.2112 | 0.1844 |
| $\text{RMSE}_m$ | 0.0238 | 0.0231 | 0.0225 | 0.0223 | 0.0208 | 0.0175 | 0.0132 |

Table 5, results from particle RMSE and grid-level $\text{RMSE}_m$ are consistent, which supports the validity and robustness of using mass RMSE in our setting.

We study whether we can address the particle misalignment issue by downsampling the reference MPM simulator to $r_p = 1/1.75$, ensuring both methods operate on identical sparse particle distributions. Under this setting, we compare two metrics: particle RMSE (our method vs. MPM at $r_p = 1/1.75$) and grid-level mass RMSE. As shown in Table 6, both metrics exhibit consistent trends across various $r_c$ values, providing additional validation for the reliability of mass-based RMSE in our experimental framework.

Table 6: Comparison of Particle RMSE and mass-based RMSE ($\text{RMSE}_m$) under different $r_c$ values for the WATER2D setup ($r_t = 2, r_p = 1/1.75$).

|  | $r_c = 0$ | $r_c = 0.3$ | $r_c = 0.5$ | $r_c = 0.6$ | $r_c = 0.7$ | $r_c = 0.8$ | $r_c = 0.9$ |
|---|---|---|---|---|---|---|---|
| Particle RMSE | 0.2755 | 0.2610 | 0.2550 | 0.2603 | 0.2375 | 0.2176 | 0.1619 |
| $\text{RMSE}_m$ | 0.0232 | 0.0223 | 0.0214 | 0.0207 | 0.0192 | 0.0169 | 0.0144 |

We also need to emphasize that "applying G2P to obtain particle acceleration at high particle resolution" requires introducing new particles. Refer to line88: `for p in x: # grid to particle (G2P)` in `mpm128.py` from the taichi-dev GitHub repository. This means that if no new particles are introduced, the number of particles in x will remain unchanged, causing misalignment between simulation particles at different resolutions.

To implement the approach of "applying G2P to obtain particle acceleration at high particle resolution," we explored two methods for introducing new particles:

1. randomly seeding new particles, and

2. using a learned point-cloud upsampler (from the pointcloud-upsampling GitHub repository).

Our experiments revealed that both methods yield excessively high particle RMSE values (Table 7). A relative RMSE greater than 1 clearly indicates that upsampling during G2P severely disrupts particle alignment and is not a viable solution.

Table 7: Particle RMSE comparison for different upsampling methods in WATER2D setup ($r_t = 2, r_p = 1/1.75$).

| Upsampling Method | Random Upsampling | Point Cloud Upsampling |
|---|---|---|
| Particle RMSE | 1.3153 | 1.2335 |

However, if we do not upsample and directly compare GNN with the ground-truth MPM with the same number of particles (i.e. particles are always aligned), the particle RMSE is much smaller. (0.2755 vs. 0.2780). This indicates that misaligned particles introduced by upsampling dominate the metric and lead to misleading evaluations. In contrast, our grid-level mass RMSE is computed directly on the simulation output, without any heuristic postprocessing or resampling, and remains stable across all tested resolutions. For this reason, we consider it a more reliable proxy for assessing cross-resolution fidelity.

**Table 8:** Comparison of recent neural physics methods.

| Dataset | Simulation Type | Method | Input | Control | Physics Correction |
|---------|-----------------|--------|-------|---------|--------------------|
| GNS (Li et al., 2018; Kumar & Vantassel, 2022; Li et al., 2023) | Neural Simulator | GNN | Particle states | ✗ | ✗ |
| MPMNet (Sharabi et al., 2024) | Hybrid Simulator | MPM + ConvLSTM | Pressure fields | ✗ | ✓ |
| NerualMPM (Han et al., 2022) | Neural Simulator | Voxelized CNN | Voxelized grid | ✗ | ✗ |
| SGNN (Ma et al., 2023) | Neural Simulator | Subequivariant GNN | Particle states | ✗ | ✗ |
| NCLaw (Viswanath et al., 2024) | Hybrid Simulator | Neural Constitutive Model + PDE | Deformation gradient | ✗ | ✓ |
| GIOROM (Viswanath et al., 2024) | Neural Simulator | GNN + Neural Fields | Velocity fields | ✗ | ✗ |
| Ours | Hybrid Simulator | GNN + MPM + Diffusion | Particle states + User action | ✓ | ✓ |

# E  MORE RESULTS

## E.1  COMPARISON WITH PREVIOUS NEURAL PHYSICS METHODS.

We have listed and compared numerous related works in Table 8, highlighting key differences between their approaches and ours, such as the use of misaligned input modalities. Among them, GIOROM (Viswanath et al., 2024) and NeuralMPM (Sharabi et al., 2024) are the most closely related to our work, as they also perform validations based on GNS (Sanchez-Gonzalez et al., 2020). In Tables 9 and 10, we further report their performance on the WATER2D and SAND2D datasets, respectively.

**Table 9:** Comparison of grid $\text{RMSE}_{\bar{m}}$ and training GPU hours with recent neural physics methods (Water2D).

| Water2D | GNS | GIOROM | NeuralMPM | Our Hybrid Solver |
|---------|-----|--------|-----------|-------------------|
| $\text{RMSE}_{\bar{m}}$ | 0.0263 | 0.0804 | 0.0829 | 0.0186 |
| GPU Hours | 17.27h | 28.37h | 17.72h | 17.27h |

**Table 10:** Comparison of grid $\text{RMSE}_{\bar{m}}$ and training GPU hours with recent neural physics methods (Sand2D).

| Sand2D | GNS | GIOROM | NeuralMPM | Our Hybrid Solver |
|--------|-----|--------|-----------|-------------------|
| $\text{RMSE}_{\bar{m}}$ | 0.0125 | 0.2175 | 0.0785 | 0.0116 |
| GPU Hours | 17.94h | 20.43h | 15.62h | 17.94h |

On the WATER2D dataset, our hybrid solver achieves an RMSE of 0.0186, which is the lowest among all compared methods. This result significantly outperforms GIOROM (0.0804) and NeuralMPM (0.0829), and also shows a marked improvement over the GNS baseline (0.0263). Similarly, on the SAND2D dataset, our method continues to demonstrate its superiority, achieving the lowest RMSE of 0.0116. Moreover, the training cost of our solver, measured in GPU hours, remains on par with the GNS baseline for both datasets. This indicates that our hybrid approach achieves a substantial increase in simulation accuracy without incurring additional training overhead. It is important to highlight that this quantitative comparison is limited to passive simulation scenarios. A direct comparison of interactive, controllable simulations was not possible, as other methods like GIOROM and NeuralMPM do not natively support user actions, a key feature of our framework.

## E.2   GRID $\text{RMSE}_{\bar{m}}$ OF FLUID SIMULATIONS OVER RANDOM SEEDS

To ensure a fairer comparison, we conducted experiments using three different random seeds. The results, as shown in Table 11, demonstrate that our hybrid solver consistently outperforms the original neural physics across all datasets.

**Table 11:** Grid $\text{RMSE}_{\bar{m}}$ of fluid simulations on different scenarios, over three random runs.

| $\text{RMSE}_{\bar{m}}$ | Water (2D) | Sand (2D) | SandRamps (2D) | WaterRamps (2D) | Water (3D) | Sand (3D) | Water-Sand (2D) |
|---|---|---|---|---|---|---|---|
| Neural Physics | 0.0263 (1.15e-6) | 0.0125 (2.59e-7) | 0.0101 (3.23e-8) | 0.0229 (2.09e-6) | 0.0048 (6.58e-7) | 0.0025 (2.11e-8) | 0.0441 (3.51e-6) |
| Our Hybrid Solver | 0.0186 (8.17e-6) | 0.0116 (6.88e-8) | 0.0096 (1.00e-9) | 0.0171 (3.16e-6) | 0.0022 (1.77e-8) | 0.0013 (1.08e-7) | 0.0149 (2.38e-6) |

## E.3   LATENCY OF FLUID CONTROLNET

For real-time performance, we report diffusion model inference time across scenes, as shown in Table 12. The latency is measured after model compilation and kernel warm-up, ensuring that initialization overhead is excluded. Further details of the latency measurement methodology are provided in Appendix C.1.

Table 12: Latency of our Fluid ControlNet.

| Dataset | Water-2D | Sand-2D | Water-3D | Sand-3D |
|---|---|---|---|---|
| Latency (ms) | 18.714 | 20.724 | 20.316 | 27.026 |

## E.4   MORE VISUALIZATIONS

**Fluid Simulations.**   Figure 14 presents the visualizations of all models discussed throughout the paper. Here, we show a comparison of intermediate frames from a single trajectory. It is evident that, due to the hybrid design of our hybrid solver, our method produces visual results that are more similar to MPM ($r_p = 1/1.75$) simulations. Since MPM ($r_p = 1/1.75$) is highly consistent with MPM (ground truth), the outputs of our Hybrid solver also align better with MPM compared to the original neural physics. This demonstrates that our approach effectively balances computational efficiency and accuracy.

**More Visualizations of Fluid Control.**   Figure 15 presents additional visualizations of generative fluid control across a variety of tasks, both 2D and 3D control signals. We can see that our approach consistently generates physically plausible and visually accurate outcomes that align closely with the target controls across all fluid types and dimensions, demonstrating strong control capability. These results further confirm the effectiveness of our method in achieving both visually appearing and physically plausible fluid control.

Despite these issues, the generated force fields still guide the fluid in the intended direction, and performance remains qualitatively acceptable. For high-precision or depth-sensitive 3D control, future work could explore 3D-aware sketching or explicit 3D conditioning. We will include this discussion in the camera-ready version to better explain current limitations and inform future improvements.

## F   LIMITATIONS

Our current limitations are: 1) The control step $T_{\text{ctl}}$ is fixed at 100 and is not adaptive to the difficulty of the control scenario; 2) Errors are introduced by the inference of neural physics at low resolution. The potential solutions are: 1) Training the diffusion-based controller to unroll different numbers of steps to adapt to challenging control scenarios; 2) Training a super-resolution model to correct errors introduced by simulating neural physics at low spatial resolution. However, addressing these limitations is beyond the scope of this paper, and we plan to study them in our immediate future work.

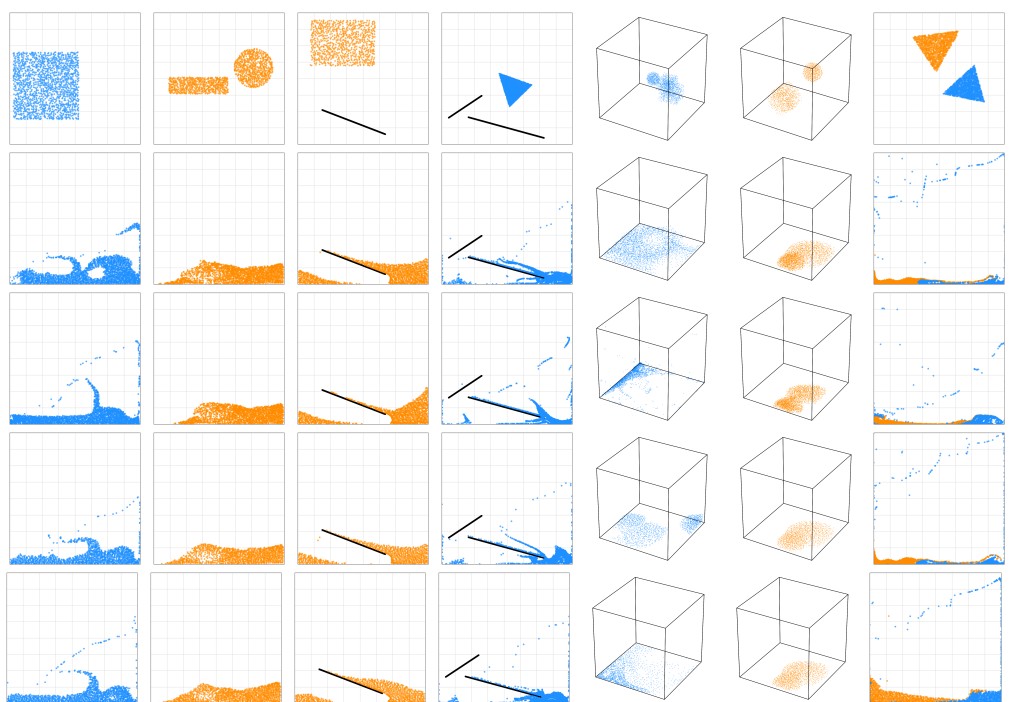

Figure 14: Visualizations of fluid simulations by different methods, over different scenarios. From left to right: Water (2D), Sand (2D), SandRamps (2D), WaterRamps (2D), Water (3D), Sand (3D), Water-Sand (2D). From top to bottom: Initial, MPM (ground truth), Original Neural Physics, MPM ($r_p = 1/1.75$), Our Hybrid Solver.

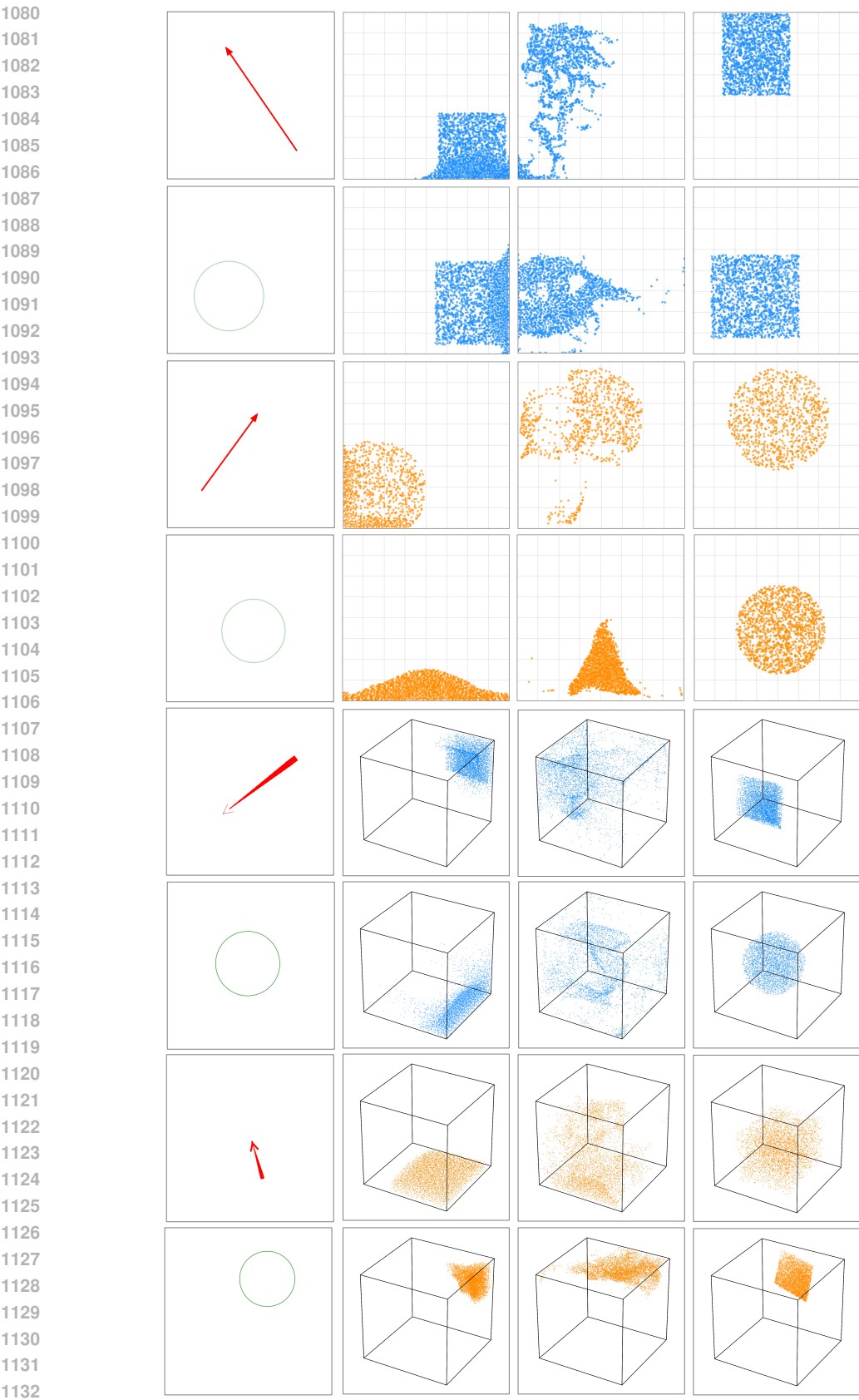

Figure 15: More visualization of generative fluid control. From top to bottom: Water2D, Sand2D, Water3D, and Sand3D, each with two types of control signals (arrows for motion direction, and oval shapes for target spatial positions). From left to right: control signal, initial, ours, ground truth.

