# OpenReview forum: "Hybrid Neural-MPM for Interactive Fluid Simulations in Real-Time"
_ICLR.cc/2026/Conference — Submitted to ICLR 2026_

### Official Review · Reviewer_iDWG · 2025-10-27

**Soundness:** 2
**Presentation:** 2
**Contribution:** 2
**Rating:** 4
**Confidence:** 2

**Summary:**

The paper proposes a fluid simulation pipeline integrating numerical simulation, neural physics and generative control. This one provides better latency than existing physics-based methodologies by only employing numerical simulation when encountering complex fluid dynamics through an automatic fallback. In particular, a GNN-based neural simulator handles low-latency updates, while a fallback to the Material Point Method (MPM) retains accuracy when the dynamics is more complex. In addition, a diffusion-based generative controller enables interactive control by mapping user sketches to external force fields. Experiments show reduced latency (11-29%) and competitive physical fidelity across 2D and 3D settings.

**Strengths:**

- The method is simple and intuitive, with the fallback mechanism ensuring that costly numerical simulation is only performed when the complexity of the fluid dynamics warrants it. While hybrid pipelines are sometimes under-appreciated in the literature, these often yield the best trade-offs.
- The latency reduction is significant. Reducing ~10-30% latency while maintaining fidelity is impressive and of practical utility for interactive graphics applications.
- Interactive control via sketches is visually compelling and, to the best of my knowledge, conceptually new.

**Weaknesses:**

- My main concern lies on the novelty of the proposed methodology with respect to PAC-Nerf [1]: although used for different objectives, both this work and Pac-NERF combine a learned neural surrogate with a physics-based simulator to get both efficiency and physical plausibility. This work should be mentioned and thoroughly discussed.
- While visually compelling, the sketch interface seems more an application layer than a core technical contribution. It’s unclear to me if the diffusion-based control module is novel or simply applied to this domain.
- The presentation needs some minor adjustment and refinement, with e.g. table 2 floating over section titles.
- While I am not up to date with all the applicable methods, the set of considered baselines seems somewhat limited. This makes it hard to assess the actual benefits with respect to the state of the art.

Considering the weaknesses and the strengths, I am inclined to reject at this time. However, as this field is not my primary area of expertise, I am happy to revise my score if these concerns are adequately addressed in the rebuttal or not shared by the other reviewers.

[1] Li, Xuan, et al. "Pac-nerf: Physics augmented continuum neural radiance fields for geometry-agnostic system identification." ICLR 2023

**Questions:**

- Can the fallback be learned or adaptive instead of rule-based? E.g. by having a confidence-based trigger instead of a fixed threshold.
- I did not fully understand what influences the latency reduction (e.g. when to expect 10% rather than 30%). Is this dependent on the complexity of the dynamics?

---

> ### Author Response · Authors · 2025-11-20
> **Rebuttal by Authors**
>
> We truly thank the time and effort of reviewer iDWG in reviewing our paper!
>
> > **Q1:** Novelty of the proposed methodology with respect to PAC-Nerf.
>
> We thank the reviewer for this comment. We respectfully distinguish our work from PAC-NeRF [1]. PAC-NeRF focuses on Inverse problems (System Identification/Geometry reconstruction from video). Our work focuses on Forward simulation and Generative control.
> We use GNNs and Diffusion for real-time interactivity, whereas PAC-NeRF uses Neural Radiance Fields for visual reconstruction. We will add a detailed discussion in the revised manuscript.
>
> > **Q2:** Diffusion-based control module is novel or simply applied to this domain?
>
> We thank the reviewer for this comment and would like to clarify the novelty of our approach. As recognized by Reviewers **XUG4** and **qrHf**, who found our method to be novel and creative. While the sketch interface is the user-facing application layer,  this is not a simple application of an existing model; its novelty lies in **how it is trained to solve this unique, ill-posed problem**.
>
> The key challenge is the lack of training data: creating paired examples of "user sketches" and the corresponding "optimal spatiotemporal force fields" to achieve them is intractable. One of our primary technical contributions is solving this data-generation problem via a novel "reversed simulation" strategy. We programmatically generate training data by first simulating a forward trajectory, then solving for the forces required to reverse that trajectory, and finally synthesizing a corresponding sketch.
>
> Therefore, our contribution is a new generative framework for fluid control and which well aligns with our proposed hybrid solver. (1) formulates the abstract "user intent-to-physics" problem as a conditional generation task and (2) introduces a novel reverse-modeling technique to create the necessary training data. This specific diffusion architecture, trained via this strategy to generate dynamic force fields for fluid control, represents a new approach in this domain.
>
> > **Q3:** Limited baselines for related work.
>
> For the neural simulator component, we provide a direct, quantitative comparison against recent state-of-the-art methods (including GNS, GIOROM, and NeuralMPM) in Appendix E (Tables 9 and 10). These results demonstrate our hybrid solver's **competitive fidelity and performance**. For our interactive generative controller, the task of real-time, sketch-based fluid control is a relatively **new domain with a scarcity of established benchmarks**. To our knowledge, no prior method tackles this specific problem of mapping freehand sketches to dynamic force fields in real-time. Therefore, to rigorously validate our method's effectiveness, we implemented a simple, intuitive baseline that is a spatiotemporal constant force field. The results in Table 3 and Figure 11  show our diffusion-based approach significantly and consistently outperforms this baseline.
>
> > **Q4:** Can the fallback be learned/adaptive?
>
> We are very grateful for the reviewer's suggestions. Currently, we use a rule-based trigger for **maximum interpretability and zero overhead**. A "confidence-based" learned trigger is possible (e.g., using the GNN's uncertainty), but estimating uncertainty in GNNs often requires ensembles (expensive). Our cosine-similarity metric acts as a lightweight proxy for this uncertainty.
>
> > **Q5:** What influences latency reduction?
>
> The latency reduction depends on the complexity of the scene. In complex scenarios such as Water-Sand 2D and simulation scenarios with obstacles, the complexity trigger activates MPM more frequently to maintain accuracy, resulting in increased latency. This adaptive behavior is designed to ensure fidelity.

---

> ### Author Response · Authors · 2025-11-26
> **Looking forward to more discussions.**
>
> Dear Reviewer iDWG:
>
> As the author-reviewer discussion period has started for a few days, we will appreciate if you could check our response to your review comments soon. This way, if you have further questions and comments, we can still reply before the author-reviewer discussion period ends. If our response resolves your concerns, we kindly ask you to consider raising the rating of our work. Thank you very much for your time and efforts.
>
> Best regards,
>
> Authors of Submission #13118

---

### Official Review · Reviewer_qrHf · 2025-10-27

**Soundness:** 1
**Presentation:** 3
**Contribution:** 2
**Rating:** 4
**Confidence:** 4

**Summary:**

This paper proposes a neural physics system for real-time, interactive fluid simulation. The work is highly innovative and engaging. It implements a system that employs an AI solver for low-complexity fluid dynamics and a traditional solver for high-complexity dynamics, augmented by a diffusion-based controller for interactive user control.

**Strengths:**

1. The research addresses a highly interesting and valuable problem, aiming to balance speed, accuracy, and interactivity, presenting a prototype of a potentially practical system.
2. The core idea is novel.
3. The method for generating training data for the controller is simple yet effective.
4. The paper is clearly written.

**Weaknesses:**

1. In Figure 7, why does the grid MSE finally decrease when transitioning from the neural physics phase to the MPM phase? Intuitively, one might expect it to increase monotonically. Could the authors provide insight into this phenomenon?
2. **Sensitivity of the Fluid Complexity Threshold (`r_c`)**: How sensitive is the hyperparameter `r_c`, used to trigger the MPM solver? If the simulation scenario changes, would the value of `r_c`require adjustment? This dependency may limit the practical utility and generalizability of the simulator.
3. **Integration of Controller and Neural Solver**: The force field learned by the Diffusion-based Fluid ControlNet must be used in conjunction with MPM. Why wasn't the interaction between the controller and the neural physics solver explored? How could such an integration be implemented?
4. **Efficiency of the Diffusion Model**: Sampling from diffusion models is often computationally intensive. What sampling strategy was employed, and what is the associated latency? Are there potential avenues for optimization?
5. **Limited Performance**: While the overall system idea is intriguing, the visual results presented in Figure 11 suggest that the network may not have effectively learned meaningful physical dynamics. Although the latency might be low, the current level of accuracy appears insufficient for practical application. How does the authors plan to bridge this significant gap between performance and usability?

**Questions:**

See Weaknesses

---

> ### Author Response · Authors · 2025-11-20
> **Rebuttal by Authors - (Part1)**
>
> We truly thank the time and effort of reviewer qrHf in reviewing our paper!
>
> > **Q1:** Figure 7: Why does Grid MSE decrease when transitioning to MPM?
>
> We agree that, in principle, one would expect the error to accumulate monotonically over longer rollouts, as is indeed observed in the black curve of Fig. 7. However, the behavior of the red curve is influenced by the fallback mechanism built into our hybrid simulator. Specifically, when the acceleration discrepancy between consecutive frames becomes too large, the system automatically falls back to the MPM solver. This fallback occurs more frequently in the later part of the rollout, allowing the MPM component to “correct” the state and prevent further drift. As a result, the accumulated error can partially decrease once the simulation enters the MPM phase more consistently. This is not a sign of instability, but evidence that the hybrid mechanism is effectively leveraging MPM to recover accuracy when the GNS is uncertain.
>
> > **Q2:** Sensitivity of the fluid complexity threshold.
>
> We conducted a sensitivity analysis for this exact parameter, presented in Figure 6(d) and Table 1. These results plot the trade-off between grid-level RMSE and simulation latency as $r_c$ is varied. As shown, this relationship forms a smooth trade-off curve rather than a sharp, unstable threshold, indicating that the system is not overly sensitive to this value. While $r_c=0.8$ is optimal for our test set, values between $0.6$ and $0.9$ still provide better latency/error trade-offs than pure MPM or pure Neural Physics. This robustness suggests that while fine-tuning $r_c$ can optimize performance for a specific scenario, the simulator's practical utility and generalizability are not limited.
>
> > **Q3:** Integration of controller and neural solver.
>
> We thank the reviewer for this insightful question. This separation is made to ensure physical fidelity and system robustness during the critical interactive control phase. Our Neural Physics Solver is a high-speed approximator. It is trained to accelerate physical dynamics (e.g., fluid falling, splashing naturally) where some approximation error is acceptable in exchange for low latency. Our MPM Solver is the high-fidelity physical engine. It is the "ground truth" that correctly handles complex interactions and external forces. When a user draws a sketch, the Diffusion Controller generates a strong, complex, and "unphysical" (in an artistic sense) external force field. The Neural Physics Solver was never trained on data containing such massive, externally-generated forces. Applying these forces to the neural solver would be a **severe out-of-distribution (OOD)** problem, and it would almost certainly produce unstable or physically nonsensical results. Conversely, the MPM solver will accept and robustly integrate any external force field. Therefore, to ensure the user's control input is met with a stable and plausible physical reaction, we must apply the control forces to the high-fidelity MPM solver.
>
> Even if we did attempt to apply the generated forces to the neural solver, this massive external intervention would immediately register as a spike in "Fluid Complexity" (an OOD state). This spike would instantly trigger our hybrid system's safeguard, which would force a fallback to the MPM solver anyway. Our approach of applying control forces only during the MPM phase is therefore the most direct and robust implementation of this logic.
>
> To achieve the deep integration the reviewer suggests, one would need to undertake a significantly more complex research task: A new, "control-aware" neural physics solver would have to be trained. This new GNN would need to be trained not just on passive dynamics, but on a massive dataset of pre-controlled simulations. Its inputs would have to be augmented to accept the (1) particle states and (2) the generated force field as conditions. This would fundamentally change the GNN's task from "predict next step" to "predict next step given this external force," likely increasing its complexity and latency, and undermining its original purpose as a lightweight accelerator for passive simulation. Our current design deliberately separates these concerns to maintain high-speed passive simulation and high-fidelity active control.

---

> ### Author Response · Authors · 2025-11-20
> **Rebuttal by Authors - (Part2)**
>
> > **Q4:** Efficiency of the diffusion model.
>
> To ensure our generative controller meets the demands of real-time interaction,  we employ a DDIM scheduler and found that we could achieve high-quality, effective force fields using only 8 denoising steps during inference. We provided a detailed latency for this in the Appendix. As reported in Table 12, the resulting inference latencies for our Fluid ControlNet are well-suited for real-time applications. These latency measurements were taken after model compilation and kernel warm-up to exclude any initialization overhead, as detailed in Appendix C.1.
>
> > **Q5:** Limited performance.
>
> We would like to clarify the specific role of Figure 11 and the division of labor in our proposed system.
> Figure 11 is not intended to evaluate the accuracy of our physics simulator (the GNN). Rather, its purpose is to demonstrate the **effectiveness of our generative controller**. It compares our diffusion-based force generation ("Ours") against a simple "spatiotemporal constant force" baseline. As shown quantitatively in Table 3, our method significantly outperforms this baseline, producing results that are much closer to the ground truth. We respectfully suggest that static images are insufficient for judging fluid dynamics; the supplementary videos provide the full, continuous sequences for Figure 11, which we believe more clearly demonstrate the stability and plausibility of the controlled motion.
>
> We do not claim that the diffusion controller itself learns the complete set of physical dynamics. This would be an extremely difficult task. Instead, our system is designed to explicitly separate artistic intent from physical execution. The Diffusion Controller's job is to learn the complex mapping from a user's sketch to a time-varying external force field. The MPM Solver's job is to be the physics engine.
>
> The "gap" the reviewer refers to is bridged by **our core design: the force field generated by the diffusion model is applied directly to the high-fidelity MPM solver**. It is the MPM solver that guarantees the final motion adheres to fluid dynamics and physical laws (like momentum and incompressibility). The generative model only needs to learn what force to suggest, and the physics engine ensures that the result of applying that force is stable and physically plausible. This design is precisely what ensures practical usability, as we are not relying on a neural network to be a perfect, all-in-one simulator.

---

> ### Author Response · Authors · 2025-11-26
> **Looking forward to more discussions.**
>
> Dear Reviewer qrHf:
>
> As the author-reviewer discussion period has started for a few days, we will appreciate if you could check our response to your review comments soon. This way, if you have further questions and comments, we can still reply before the author-reviewer discussion period ends. If our response resolves your concerns, we kindly ask you to consider raising the rating of our work. Thank you very much for your time and efforts.
>
> Best regards,
>
> Authors of Submission #13118

---

### Official Review · Reviewer_XUG4 · 2025-11-01

**Soundness:** 3
**Presentation:** 1
**Contribution:** 2
**Rating:** 2
**Confidence:** 4

**Summary:**

This paper presents a hybrid neural Material Point Method (MPM) framework that combines physically based simulation with learned material models to improve efficiency and offer artistic. A graph neural network (GNN) predicts particle accelerations within an MPM solver, operating on lower resolutions to cut latency while remaining compatible with the standard p2g/g2p update loop. The system includes a safeguard: it monitors rollout drift via a complexity proxy and, when a threshold is crossed, automatically falls back to a classical MPM step to correct errors. Lastly, the authors introduce a diffusion-based “Fluid ControlNet” that learns to generate external force/acceleration fields from user sketches; training targets are obtained by a reverse-simulation paradigm that solves the accelerations needed to invert forward dynamics. Results demonstrate a few 2D/3D water/sand scenes.

**Strengths:**

- The proposed fallback mechanism is am elegant safeguard against simulation drift, enabling stability against potential issues encountered in long-horizon rollouts.

- Integrating a diffusion model to learn reverse accelerations for artistic control is a creative idea, offering a novel way to steer physically based simulations through generative methods.

**Weaknesses:**

- The presentation in its current format is sub-par. Several plots rely on scattered points that difficult clear understanding, while Figure 7 is particularly confusing. Why does the red rollout abruptly stop? It's also suspicious that the hybrid error high before the cutoff, so it seems like the approximation has a high baseline error. Moreover the abrupt truncation of the red plot could be an attempt of potentially masking issues of the method on further roll-outs.

- The related-work discussion is brief relative to the breadth of prior neural simulators, hybrid solvers, and control methods; important positioning (what’s new vs. NeuralMPM/MPMNet/Neural SPH) could be sharper.

- No supplemental video is provided, which is crucial for judging visual fidelity, stability, and interactive behavior of fluids (especially the control sequences).

- Results are limited and could be better presented. Is hard to understand what Figure 11 is demonstrating beyond qualitative snapshots,

- The paper motivates diffusion models by claiming “strong conditional generation with temporal coherence and spatial flexibility,” but the architecture description does not employ explicit temporal attention or other mechanisms typically used to enforce coherence over long horizons in diffusion video models; this mismatch deserves either justification or an ablation.

- The scope feels split across its two contributions: real-time acceleration via a neural–MPM hybrid with fallback and sketch-driven “artistic” control via a diffusion controller. However it feels like both contributions are under-explored, seems like each direction could merit a focused paper with deeper ablations.

**Questions:**

- What are the parameters used to model the sand and water materials?
- Did the authors collect feedback from users about how intuitive/efficient the proposed force-based control strategy is?

---

> ### Author Response · Authors · 2025-11-20
> **Rebuttal by Authors - (Part1)**
>
> We truly thank the time and effort of reviewer XUG4 in reviewing our paper!
>
> > **Q1:** More explanations about Figure 7
>
> In Fig. 7, the horizontal axis represents simulation time rather than the number of steps. Therefore, the red and black curves have different lengths because they correspond to the same number of simulation steps but not the same total simulated time. This is **not a cutoff** or truncation.
> Regarding the initially higher error of the red curve: the red trajectory is produced by the accelerated version of GNS. As is commonly observed, the accelerated variant tends to exhibit a larger initial prediction error compared with the baseline GNS. This explains the higher starting error and is not related to any masking of performance at longer rollouts.
> We promise to include more annotations into this figure and its caption for better illustration.
>
> > **Q2:** Related work discussion, positioning of our work.
>
> We thank the reviewer for this suggestion. We will expand this discussion in the final version. We wish to clarify that our primary contribution is not just an alternative hybrid solver, but rather the first complete framework for real-time, sketch-based interactive control.
>
> While methods like NeuralMPM, MPMNet, and Neural SPH have made significant advances in accelerating simulations, their primary focus is not on user interaction. Our work is fundamentally differentiated by two key, synergistic contributions:
>
> - **Hybrid Solver for Interaction**: Our hybrid solver (GNN + MPM fallback ) is specifically designed and ablated (Figure 6) to serve a particular goal: providing the low-latency, high-fidelity engine required for a practical interactive loop.
> - **Novel Interactive Control**: The main novelty is our diffusion-based generative controller. This component, trained via a novel reverse-simulation strategy, directly addresses the interactive control task by interpreting user sketches.
>
> We did provide a direct quantitative comparison of the neural simulators against our method in Appendix E (Tables 8, 9, 10), demonstrating our solver's competitive fidelity. However, our paper's central novelty lies in the synthesis of this real-time solver with the generative controller to enable a new capability: practical, real-time, artist-friendly fluid control.
>
> We promise to include discussions on these comparisons in our camera ready.
>
> > **Q3:** Supplemental videos and more results
>
> We would like to respectfully clarify that we **indeed have included a comprehensive set of video results** in the supplementary material.
>
> About Figure 11, the supplement includes the full, continuous video sequences corresponding to these exact qualitative snapshots. Furthermore, the supplementary material contains numerous additional results, including videos for various materials (like Water and Sand) in both 2D and 3D scenarios, more examples of our control sequences, and complete demonstrations of our full pipeline integrating the hybrid simulation with the interactive fluid control. We kindly hope the reviewer to view these materials, as we are confident they provide the necessary evidence to fully assess our contributions.
>
> > **Q4:** No employ explicit temporal attention in Diffusion.
>
> We thank the reviewer for this insightful question. Our hybrid framework addresses temporal coherence through a different, more physically grounded approach.
>
> Our model is conditioned on an **implicit temporal context**. The input to the Fluid ControlNet is not a single static frame, but rather the particle states (positions and velocities) from the 6 preceding timesteps. This local history, combined with the embedding of the current control timestep, provides sufficient context for the model to predict the instantaneous force field required at the present moment, rather than needing to model the entire future trajectory.
>
> More fundamentally, long-range temporal coherence is explicitly **enforced by integrating our neural physics** with the numerical MPM solver, which guarantees the physical correctness of fluid dynamics after the control.
> Our generative model is not responsible to predict subsequent particle states or video frames, but to generate a spatiotemporal acceleration field (i.e., a force field). This force field is then applied to the MPM simulation, and it is the physics engine itself that integrates these forces over time to update particle velocities and positions. The MPM solver is inherently stateful and ensures temporal continuity by conserving momentum and adhering to physical laws.

---

> ### Author Response · Authors · 2025-11-20
> **Rebuttal by Authors - (Part2)**
>
> > **Q5:** Scope split across its two contributions, and each direction could merit a focused paper with deeper ablations.
>
> We respectfully argue that these two contributions are ***not split***, but rather are two essential and deeply synergistic components of a single, unified goal: achieving practical, real-time interactive fluid control.
>
> The first contribution, our Hybrid Neural-MPM solver, serves as the foundational engine. The primary bottleneck for any interactive system is latency. A generative controller, no matter how fast, is rendered impractical if the underlying physics simulation cannot keep pace. Our hybrid solver directly tackles this simulation bottleneck, providing the low-latency, high-fidelity platform that is an absolute prerequisite for any meaningful real-time interaction.
>
> The second contribution, the Fluid ControlNet, makes our work not merely to accelerate passive simulations, but to "close the loop" with the user, enabling them to intuitively manipulate the fluid. Our sketch-based controller provides this artist-friendly, interactive interface. This mechanism demonstrates the purpose of the real-time engine, transforming it from a passive viewer into a responsive, controllable system.
>
> Therefore, one contribution cannot be fully realized without the other; the hybrid solver enables the interaction, and the interactive controller gives meaning to the real-time acceleration. We believe our ablations (Figure 6 for the hybrid solver's trade-offs and Table 3/Figure 11 for the controller's effectiveness) are sufficient to validate this primary, unified contribution, which is the complete, end-to-end framework for real-time interactive control.
>
> > **Q6:** What are the parameters used to model the sand and water materials?
>
> The physical parameters in MPM simulation are as follows: For Sand, we set the density ($\rho$) to $400$. The elastic properties are defined by a Young's Modulus ($E$) of $4.0 \times 10^5$ and a Poisson's Ratio ($\nu$) of $0.25$, from which the Lame parameters ($\mu$, $\lambda$) are derived. The friction coefficient (mu_b) is $1.0$, and the hardening model parameters are $h_0=35$, $h_1=9$, $h_2=0.2$, and $h_3=10$. For Water, we use a density ($\rho$) of $1.0$ and a Young's Modulus ($E$) of $400$ to govern its incompressibility. Gravity is set to $9.8$. We will add a detailed parameter table to the Appendix.
>
> > **Q7:** Feedback from users about fluid control results?
>
> We thank the reviewer for this valuable suggestion.
>
> We agree that user feedback will be a useful evaluation for our interactive system, and we will try to include a user study in our camera ready.

---

> ### Author Response · Authors · 2025-11-26
> **Looking forward to more discussions.**
>
> Dear Reviewer XUG4:
>
> As the author-reviewer discussion period has started for a few days, we will appreciate if you could check our response to your review comments soon. This way, if you have further questions and comments, we can still reply before the author-reviewer discussion period ends. If our response resolves your concerns, we kindly ask you to consider raising the rating of our work. Thank you very much for your time and efforts.
>
> Best regards,
>
> Authors of Submission #13118

---

> > ### Comment · Reviewer_XUG4 · 2025-11-27
> >
> > I still feel that the changes needed for the paper to be accepted at ICLR would be too many, and thus would require another revision cycle. I appreciate the authors rebuttal, but I'm keeping my score.

---

### Official Review · Reviewer_xowZ · 2025-11-04

**Soundness:** 1
**Presentation:** 3
**Contribution:** 2
**Rating:** 2
**Confidence:** 4

**Summary:**

This paper proposes a hybrid neural-numerical framework for real-time, interactive fluid simulation. A graph-neural-network  runs at reduced spatio-temporal resolution to cut latency, while a safeguard falls back to an MPM solver to preserve fidelity when particle acceleration is high. On top of simulation, the authors introduce a diffusion-based controller (Fluid ControlNet) trained via a reverse-simulation data pipeline to generate external force fields from user freehand sketches for intuitive interactive fluid control.

**Strengths:**

The blend of classical and neural approaches effectively balances their respective strengths and limitations, making this a pragmatic and promising path forward.

**Weaknesses:**

While the hybrid direction is practical, I find the core mechanism of monitoring the error then hard-switching to a classical solver is a generic wrapper with limited novelty. As presented, it could be applied to most neural simulators; the paper should demonstrate if any part of their design makes fallback uniquely efficient, and compare against to other neural simulation baselines by applying the same fallback onto other neural simulators.

The generative controller is also weakly motivated: classic optimization-based methods for interactive fluid control (e.g., “Fluid Control Using the Adjoint Method” ; McNamara et al., 2004) tackle the same task without needing diffusion models for control. Table 3 also showed little improvement by using the generative model.

**Questions:**

Novelty of fallback. What, concretely, makes your fallback uniquely efficient versus a generic wrapper, if there is any? Can you support it with an experiment?

Comparison with baselines: please compare your method with other neural simulation methods by applying the same fallback mechanism so performance on the neural simulation can be evaluated.

Controller motivation & baselines. Why diffusion over classic optimization/adjoint control (e.g., McNamara et al., 2004) for the same interactive tasks? Table 3 shows little evidence that the generative force field is helping much.

Discussion with related works:   McNamara et al., 2004 is closely related and is not discussed in related works. Please check missing references.

**Details Of Ethics Concerns:**

None.

---

> ### Author Response · Authors · 2025-11-20
> **Rebuttal by Authors**
>
> We truly thank the time and effort of reviewer xowZ in reviewing our paper!
>
> > **Q1:** Novelty and efficiency of the fallback mechanism?
>
> 1) Novelty. As Review **XUG4** raised, "the proposed fallback mechanism is an elegant safeguard". To our best knowledge, we are the **first to automate the hybridization between MPM and neural physics with their trade-off carefully studied**. The most relevant work is "Krishna Kumar and Yonjin Choi. Accelerating particle and fluid simulations with differentiable graph networks for solving forward and inverse problems. 2023", where they studied interleave simulations by MPM and neural physics, but their switch was hand-tuned without any automation, and unlike our work, they also did not carefully study the trade-off between efficiency and simulation errors.
> 2) Efficiency. Our fallback mechanism is highly efficient because the trigger relies solely on quantities already computed by the neural physics model, specifically, the per-particle acceleration vectors.
>    By evaluating the cosine similarity of these accelerations over a short temporal window, we obtain an early-warning signal that is strongly negatively correlated with future rollout error (Figure 5).
>    Crucially, this requires no additional neighbour search, grid interpolation, or P2G operations, making the monitoring overhead negligible and significantly lower than alternative criteria such as velocity-divergence or neighbour-density checks (L.135–138).
>
> > **Q2:** Comparison with baselines.
>
> We include rich comparisons with other baselines in Table 9 and 10.
> We observe that GNS *already largely outperforms other baselines*, yielding the lowest error among learned simulators.
> By adopting our hybrid neural physics on top of GNS, we can further outperform GNS.
> This makes further adoption on other baselines less meaningful, because we **mainly focus on improving MPM** (our method is named as "HybridMPM"), and *never claim the universality of our hybridization as our contribution*.
>
> > **Q3:** Controller motivation & baselines?
>
> We thank the reviewer for this observation and would like to clarify the fundamental difference in the task and motivation against McNamara et al., 2004.
> We acknowledge that McNamara et al., 2004 share our objective of directing fluid motion, and we will include a detailed discussion in our camera ready.
> However, classic optimization-based methods [1], are fundamentally **offline systems**.
> They are to solve a complex optimization problem: finding the optimal forces required to match a set of pre-defined, precise keyframes. This iterative optimization process is computationally prohibitive for interactive use, often requiring a lot of time to compute a result.
> Our work targets **interactive fluid control**. The motivation for our generative controller is to interpret user-friendly, freehand sketches, which represent high-level 'intent' rather than precise target states, and instantaneously generate a physically plausible, dynamic force field. This is a generative learning task.
>
> Regarding Table 3, it demonstrates that our diffusion-based model achieves a consistent quantitative improvement over the baseline in all tested scenarios. This baseline is a "spatiotemporal constant force field", whereas our model successfully learns to generate a complex, time-varying field. This improvement validates that our diffusion-based approach successfully learns this non-trivial mapping from sketch-based intent to an effective, dynamic control field, which ultimately better aligns the fluid motion with the user's sketch.
>
> [1] McNamara, Antoine, et al. "Fluid control using the adjoint method." 2004

---

> ### Author Response · Authors · 2025-11-26
> **Looking forward to more discussions.**
>
> Dear Reviewer xowZ:
>
> As the author-reviewer discussion period has started for a few days, we will appreciate if you could check our response to your review comments soon. This way, if you have further questions and comments, we can still reply before the author-reviewer discussion period ends. If our response resolves your concerns, we kindly ask you to consider raising the rating of our work. Thank you very much for your time and efforts.
>
> Best regards,
>
> Authors of Submission #13118

---

### Author Response · Authors · 2025-11-29
**Rebuttal Summary  - (Part1)**

Dear PC, AC, and reviewers:

Since further public discussions are no longer allowed, we would like to post a rebuttal summary here to note all the comments and our response. We sincerely appreciate all the reviewers (**xowZ, XUG4, qrHf, iDWG**) for their thoughtful suggestions that help improve the paper quality. We are encouraged that the reviewers unanimously recognized the value of our work in several key aspects:

- **Innovation & Technique Contribution**: Reviewers highlighted the "creative idea" (XUG4) and "innovative" nature (qrHf) of our core concepts. Specifically, the integration of diffusion for reverse-simulation control was praised as "conceptually new" (iDWG) and offering a "novel way to steer physically based simulations" (XUG4).
- **Practicality & Efficiency**: The trade-off between speed and fidelity was well-received. Reviewers noted the "significant latency reduction" (iDWG), described the approach as "pragmatic and promising" (xowZ), and "prototype of a potentially practical system" (qrHf).
- **Hybrid Design**: The hybrid solver architecture was commended as an "elegant safeguard" (XUG4) that yields the "best trade-offs" (iDWG) by effectively balancing classical and neural strengths (xowZ).

However, we also address the major concerns.

> Novelty and Specificity of the Fallback Mechanism (xowZ)

We emphasize that we are the **first to automate the hybridization between MPM and neural physics with a carefully studied trade-off** (Fig. 6). Unlike generic switching, our fallback trigger is highly specific and efficient: it relies on the cosine similarity of per-particle acceleration vectors. As shown in our rebuttal [[xowZ Q1](https://openreview.net/forum?id=6vX0LH9Yt7&referrer=%5BAuthor%20Console%5D(%2Fgroup%3Fid%3DICLR.cc%2F2026%2FConference%2FAuthors%23your-submissions)#:~:text=Q1%3A%20Novelty%20and%20efficiency%20of%20the%20fallback%20mechanism%3F)], this metric acts as an early-warning signal strongly correlated with future rollout errors (Fig. 5) but requires zero additional overhead (no neighbor search or grid operations), unlike alternative criteria such as velocity divergence.

> Motivation for Generative Controller & Comparison to Optimization (xowZ)

- Offline vs. Interactive: Classic optimization methods (e.g., McNamara et al., 2004) are fundamentally offline systems designed to solve complex optimization problems for precise keyframes. This iterative process is computationally prohibitive for real-time interaction.[[xowZ Q3](https://openreview.net/forum?id=6vX0LH9Yt7&referrer=%5BAuthor%20Console%5D(%2Fgroup%3Fid%3DICLR.cc%2F2026%2FConference%2FAuthors%23your-submissions)#:~:text=Q3%3A%20Controller%20motivation%20%26%20baselines%3F)]
- Motivation: Our work targets interactive fluid control. The motivation for our generative controller is to **interpret user-friendly, freehand sketches**, which represent high-level '**intent**' rather than precise target states, and instantaneously generate a physically plausible, dynamic force field. [[xowZ Q3](https://openreview.net/forum?id=6vX0LH9Yt7&referrer=%5BAuthor%20Console%5D(%2Fgroup%3Fid%3DICLR.cc%2F2026%2FConference%2FAuthors%23your-submissions)#:~:text=Q3%3A%20Controller%20motivation%20%26%20baselines%3F)]
- Quantitative Effectiveness: Regarding Table 3, this baseline is a "spatiotemporal constant force field", whereas our model successfully learns to generate a complex, time-varying field. This improvement validates that our diffusion-based approach successfully learns this non-trivial mapping from sketch-based intent to an effective, dynamic control field, which ultimately better aligns the fluid motion with the user's sketch. [[xowZ Q3](https://openreview.net/forum?id=6vX0LH9Yt7&referrer=%5BAuthor%20Console%5D(%2Fgroup%3Fid%3DICLR.cc%2F2026%2FConference%2FAuthors%23your-submissions)#:~:text=Q3%3A%20Controller%20motivation%20%26%20baselines%3F)]

> Comparison with Baselines (PAC-NeRF, Neural Simulators) (iDWG, xowZ)

We distinguished our work (Simulation & Generative Control) from PAC-NeRF (Inverse System Identification/Reconstruction). As shown Table 9 and 10 in the Appendix, directly comparing our method against GNS, NeuralMPM, and GIOROM. Results show our hybrid solver achieves the lowest error among learned simulators while maintaining real-time performance.[[xowZ Q2](https://openreview.net/forum?id=6vX0LH9Yt7&referrer=%5BAuthor%20Console%5D(%2Fgroup%3Fid%3DICLR.cc%2F2026%2FConference%2FAuthors%23your-submissions)#:~:text=Q2%3A%20Comparison%20with%20baselines.), [IDWG Q1](https://openreview.net/forum?id=6vX0LH9Yt7&referrer=%5BAuthor%20Console%5D(/group%253Fid=ICLR.cc/2026/Conference/Authors%2523your-submissions)#:~:text=Q1%3A%20Novelty%20of%20the%20proposed%20methodology%20with%20respect%20to%20PAC%2DNerf.), [IDWG Q3](https://openreview.net/forum?id=6vX0LH9Yt7&referrer=%5BAuthor%20Console%5D(%2Fgroup%3Fid%3DICLR.cc%2F2026%2FConference%2FAuthors%23your-submissions)#:~:text=Q3%3A%20Limited%20baselines%20for%20related%20work.)]

---

> ### Author Response · Authors · 2025-11-29
> **Rebuttal Summary - (Part2)**
>
> > Clarifications on Simulation Stability (Fig. 7) and Integration (XUG4, qrHf)
>
> * Fig. 7 Dynamics: We clarified that the "drop" in error is not an artifact but explicitly demonstrates the fallback mechanism in action that the MPM solver "corrects" the drift accumulated by the neural solver. [[XUG4 Q1](https://openreview.net/forum?id=6vX0LH9Yt7&referrer=%5BAuthor%20Console%5D(%2Fgroup%3Fid%3DICLR.cc%2F2026%2FConference%2FAuthors%23your-submissions)#:~:text=Q1%3A%20More%20explanations%20about%20Figure%207), [qrHf Q1](https://openreview.net/forum?id=6vX0LH9Yt7&referrer=%5BAuthor%20Console%5D(%2Fgroup%3Fid%3DICLR.cc%2F2026%2FConference%2FAuthors%23your-submissions)#:~:text=Q1%3A%20Figure%207%3A%20Why%20does%20Grid%20MSE%20decrease%20when%20transitioning%20to%20MPM%3F)]
> * Controller Integration: We explained that applying strong external control forces to the neural solver would be Out-Of-Distribution (OOD). By applying these forces to the robust MPM solver during the fallback phase, we ensure physical plausibility without destabilizing the neural component. [[qrHf Q3](https://openreview.net/forum?id=6vX0LH9Yt7&referrer=%5BAuthor%20Console%5D(%2Fgroup%3Fid%3DICLR.cc%2F2026%2FConference%2FAuthors%23your-submissions)#:~:text=Q3%3A%20Integration%20of%20controller%20and%20neural%20solver.)]
>
> > Results, Videos, and Performance Limitations (XUG4, qrHf)
>
> We confirmed that we **indeed included a comprehensive set of video results** in the supplementary material. This includes continuous sequences for Figure 11, various material types (Water/Sand), and full pipeline demos, which are necessary to fully assess visual fidelity and stability. We clarified that Figure 11 evaluates the **controller's effectiveness** (comparing against a constant force baseline), not the simulator's physics. We do not claim that the diffusion controller itself learns the complete set of physical dynamics. This would be an extremely difficult task. Instead, our system is designed to explicitly separate artistic intent from physical execution. The Diffusion Controller's job is to learn the complex mapping from a user's sketch to a time-varying external force field. The MPM Solver's job is to be the physics engine.  This design is precisely what ensures practical usability, as we are not relying on a neural network to be a perfect, all-in-one simulator. [[XUG4 Q3](https://openreview.net/forum?id=6vX0LH9Yt7&referrer=%5BAuthor%20Console%5D(%2Fgroup%3Fid%3DICLR.cc%2F2026%2FConference%2FAuthors%23your-submissions)#:~:text=Q3%3A%20Supplemental%20videos%20and%20more%20results), [qrHf Q5](https://openreview.net/forum?id=6vX0LH9Yt7&referrer=%5BAuthor%20Console%5D(%2Fgroup%3Fid%3DICLR.cc%2F2026%2FConference%2FAuthors%23your-submissions)#:~:text=Q5%3A%20Limited%20performance.)]

---

> ### Author Response · Authors · 2025-11-29
> **Rebuttal Summary - (Part3)**
>
> > For additional explanations or justifications, we direct to the rebuttal and discussions with reviewers:
>
> * Clarification on the unified scope and synergy of contributions: [[XUG4 Q5](https://openreview.net/forum?id=6vX0LH9Yt7&referrer=%5BAuthor%20Console%5D(%2Fgroup%3Fid%3DICLR.cc%2F2026%2FConference%2FAuthors%23your-submissions)#:~:text=Q5%3A%20Scope%20split%20across%20its%20two%20contributions%2C%20and%20each%20direction%20could%20merit%20a%20focused%20paper%20with%20deeper%20ablations.)]
> * Sensitivity analysis of the fluid complexity threshold ($r_c$): [[qrHf Q2](https://openreview.net/forum?id=6vX0LH9Yt7&referrer=%5BAuthor%20Console%5D(/group%253Fid=ICLR.cc/2026/Conference/Authors%2523your-submissions)#:~:text=Q2%3A%20Sensitivity%20of%20the%20fluid%20complexity%20threshold.)]
> * Justification for no explicit temporal attention in Diffusion: [[XUG4 Q4](https://openreview.net/forum?id=6vX0LH9Yt7&referrer=%5BAuthor%20Console%5D(/group%253Fid=ICLR.cc/2026/Conference/Authors%2523your-submissions)#:~:text=Q4%3A%20No%20employ%20explicit%20temporal%20attention%20in%20Diffusion.)]
> * Efficiency/Latency details of the Diffusion model: [[qrHf Q4](https://openreview.net/forum?id=6vX0LH9Yt7&referrer=%5BAuthor%20Console%5D(/group%253Fid=ICLR.cc/2026/Conference/Authors%2523your-submissions)#:~:text=Q4%3A%20Efficiency%20of%20the%20diffusion%20model.)]
> * Novelty of Diffusion application in this domain: [[iDWG Q2](https://openreview.net/forum?id=6vX0LH9Yt7&referrer=%5BAuthor%20Console%5D(/group%253Fid=ICLR.cc/2026/Conference/Authors%2523your-submissions)#:~:text=Q2%3A%20Diffusion%2Dbased%20control%20module%20is%20novel%20or%20simply%20applied%20to%20this%20domain%3F)]
> * Physical parameters used for Sand/Water modeling: [[XUG4 Q6](https://openreview.net/forum?id=6vX0LH9Yt7&referrer=%5BAuthor%20Console%5D(/group%253Fid=ICLR.cc/2026/Conference/Authors%2523your-submissions)#:~:text=Q6%3A%20What%20are%20the%20parameters%20used%20to%20model%20the%20sand%20and%20water%20materials%3F)]
>
> In general, we believe our rebuttal demonstrates that our contributions are not merely a wrapper, but a cohesive, novel framework solving the specific challenges of real-time interactivity. By enabling automated fidelity safeguards and sketch-based generative control, we bridge the gap between passive simulation and interactive creation. Thank you very much for your time and effort.
>
> Best regards,
>
> Authors of Submission #13118

---

### Meta-Review · Area_Chair_g77D · 2026-01-05

**Summary:**

The reviewers raised concerns about novelty, scope, and experimental validation. Reviewer xowZ (score: 2) questioned whether the fallback mechanism is "a generic wrapper that most neural simulators" could use and was unconvinced by the diffusion-based controller's motivation over classical optimization methods. Reviewer XUG4 (score: 2) criticized the presentation quality, felt "the scope feels split across its two contributions," and suggested each direction "could merit a focused paper with deeper ablations." Reviewer qrHf (score: 4) appreciated the innovation but questioned threshold sensitivity and whether "the current level of accuracy appears insufficient for practical application." Reviewer iDWG (score: 4) questioned novelty relative to PAC-NeRF and found the baselines "somewhat limited."

**Reviewer Concerns:**

The authors adequately clarified that supplementary videos were provided, explained the efficiency of their cosine-similarity-based fallback trigger, provided comparisons with GNS, NeuralMPM, and GIOROM (Tables 9-10), and distinguished their work from PAC-NeRF's inverse identification task. However, key concerns remain. The novelty of the fallback mechanism is unclear: the authors acknowledge this is not universally applicable to other neural simulators. The motivation for diffusion-based control over classical optimization is weak; the authors claim classical methods are "computationally prohibitive for real-time interaction" but provide no timing comparisons. The quantitative improvement in Table 3 is marginal. The concern about split scope was not resolved; while the authors argue the components are "synergistic," this does not address whether each component has sufficient depth for a full ICLR contribution.

**Reviewer Scores:**

Reviewer xowZ would likely maintain score 2, as their core concerns about the fallback mechanism and weak controller motivation were not convincingly addressed.

Reviewer XUG4 explicitly stated "the changes needed for the paper to be accepted at ICLR would be too many," indicating they would maintain score 2.

Reviewer qrHf might increase from 4 to 6, as technical explanations for threshold sensitivity and controller integration were reasonable, though practical accuracy concerns remain.

Reviewer iDWG would likely remain at 4, as the PAC-NeRF distinction was clarified but concerns about limited baselines persist.

Average hypothetical score: 3.5. Recommendation: reject.

---

### Decision · Program_Chairs · 2026-01-26

Reject